# How Predictable Are Large Language Model Capabilities?
# A Case Study on BIG-bench

**Qinyuan Ye     Harvey Yiyun Fu     Xiang Ren     Robin Jia**
University of Southern California, Los Angeles, CA, USA
{qinyuany, harveyfu, xiangren, robinjia}@usc.edu

## Abstract

We investigate the predictability of large language model (LLM) capabilities: given records of past experiments using different model families, numbers of parameters, tasks, and numbers of in-context examples, can we accurately predict LLM performance on new experiment configurations? Answering this question has practical implications for LLM users (*e.g.*, deciding which models to try), developers (*e.g.*, prioritizing evaluation on representative tasks), and the research community (*e.g.*, identifying hard-to-predict capabilities that warrant further investigation).

We study the performance prediction problem on experiment records from BIG-bench. On a random train-test split, an MLP-based predictor achieves an $R^2$ score greater than 95%, indicating the presence of learnable patterns within the experiment records. We then formulate the problem of searching for "small-bench," an informative subset of BIG-bench tasks from which the performance on the full set can be maximally recovered. We find a subset as informative as BIG-bench Hard for evaluating new model families, while being $3\times$ smaller. Additionally, we find competitive subsets by clustering task representations learned by our MLP-based predictor and selecting tasks close to cluster centroids, highlighting the importance of task diversity in constructing "small-bench."[1]

## 1   Introduction

Large language models (LLMs) have revolutionized natural language processing (NLP) research. Typically, when researchers introduce a new set of LLMs, they release them in various sizes and conduct extensive evaluation on different tasks, while also considering different experiment configurations, such as prompting strategies and the number of in-context examples (Black et al., 2021; Zhang

---

[1]Code can be found at https://github.com/INK-USC/predicting-big-bench.

| Model Family | # param | Task | # shot | Perf. |
|---|---|---|---|---|
| GPT-3 | 3B | strategy_qa | 0 | 0.48 |
| BIG-G T=1 | 8B | elementary_math | 3 | 0.19 |
| PaLM | 64B | code_line_desc | 2 | 0.23 |
| GPT-3 | 6B | elementary_math | 1 | ? |

**How *predictable* are LLM capabilities?**

**How to evaluate new models *within budget constraints*?**

Figure 1: **Overview.** We study the problem of (1) predicting LLM performance on new experiment configurations; (2) searching for a subset of tasks which is most informative for predicting performance on remaining tasks when evaluating a new model family.

et al., 2022; Touvron et al., 2023). Given the combinatorially large space of possible experimental configurations, running all possible experiments for a new set of LLMs is impractical. This begets a critical question: to what extent can we predict the capabilities of an LLM in a given experimental setting?

Studying this problem helps address various practical issues. For LLM users, a performance prediction model could offer guidance for experiment design and decision-making by answering questions such as, "What model scale and how many shots are necessary to attain satisfactory performance for my task?" For LLM developers and the research community, a performance prediction model could lead to insights into LLM capabilities by identifying which capabilities are hard-to-predict and require further investigation, and which capabilities are highly correlated and may be deprioritized during evaluation to save budget.

We investigate the predictability of LLM capabilities on the BIG-bench (Srivastava et al., 2023) evaluation suite, as it includes a vast collection of experiment records. BIG-bench is a collaborative initiative aimed at "prob[ing] large language models and extrapolat[ing] their future capabilities." It has extensively evaluated various state-of-the-art LLMs on a diverse set of tasks contributed by the community. We gather and carefully filter these records, yielding a total of 56k+ records which we use as the "dataset" for our analysis.

We first formulate the problem of performance prediction given experiment configurations such as model family, model scale, task, and the number of in-context examples used. We compare various matrix completion, tree-based, and neural network methods in a random train-test split scenario (§3). Further, we design and experiment with various data splits, representing different types of generalization challenges, to simulate practical situations researchers may face (§4).

We then consider the problem of searching for "small-bench," a compact and informative subset of BIG-bench. This subset should allow for maximum recovery of performance on the complete BIG-bench, enabling efficient evaluation of new LLMs while maintaining evaluation generality. We formulate this as a subset search problem (§5) and empirically compare various search methods, clustering-based subset construction methods, along with widely-adopted subsets such as BIG-bench Lite and BIG-bench Hard (Suzgun et al., 2023).

Our key findings are summarized as follow:

1. LLMs' performance on BIG-bench follows predictable patterns. In the default random train-test split scenario, our best predictor, an MLP model, achieves an RMSE lower than $0.05$ (*i.e.*, on average mis-predict by $< 0.05$ when the range is $[0, 1]$) and an $R^2$ greater than $95\%$ (*i.e.*, explains more than $95\%$ variance in the target variable).

2. The predictor's performance is dependent on the assumptions of the train-test distribution. In a more challenging setting where we hold out the Cartesian product of complete model families (all model scales) and complete tasks (all numbers of shots), the predictor's performance decreases ($R^2 : 95\% \rightarrow 86\%$).

3. Performance of emergent tasks (Wei et al., 2022a) is not entirely unpredictable. In general, performance of emergent tasks is harder to predict than that of non-emergent tasks. In

specific scenarios (*e.g.*, when a related emergent task is present in the training set) our model can accurately predict emergent abilities.

4. BIG-bench Lite and BIG-bench Hard (Suzgun et al., 2023), two subsets of BIG-bench commonly used for evaluating new models, are suboptimal if the goal is to recover the performance on remaining tasks. We are able to find a subset that is as informative as BIG-bench Hard while being $3\times$ smaller by using randomized search.

5. Task diversity and task value are critical factors in constructing "small-bench." By clustering task representations learned by the MLP-based predictor and selecting tasks close to cluster centroids, we obtain competitive "small-bench" candidates. This strategy is further improved by incorporating task value information.

## 2 Related Work

**Scaling Laws and Emergent Abilities.** Pretraining scale is critical to language model capabilities. Research on scaling laws (Kaplan et al., 2020; Rae et al., 2021; Hoffmann et al., 2022) aims to categorize the relationship between pre-training compute, corpus size, model size and the test log-likelihood loss. Our work can be loosely considered as an extension to scaling laws, with three notable distinctions: (1) we focus on predicting downstream task performance; (2) we use model scale along with other experiment configuration information; (3) we mainly experiment with machine learning methods instead of explicit power laws. In this same vein, recent work has studied the effect of scale in a "pre-train then fine-tune" paradigm (Tay et al., 2022) and has explored non-monotonic scaling laws for complex scaling behaviors (Caballero et al., 2023). Another important observation about scale is that very large language models exhibit emergent abilities (Wei et al., 2022a), which are described as "unpredictable." In this work we empirically examine this claim and quantify the prediction errors under various assumptions.

**Benchmarking for LLMs.** Along with the development and scaling of LLMs, there are continuing efforts to create benchmarks that assess the capabilities of these models. One general trend for these benchmarks is transitioning from single-task (Bowman et al., 2015; Rajpurkar et al., 2016), to multi-task (Wang et al., 2018, 2019), and finally to massively multi-task (Hendrycks et al., 2021; Srivastava et al., 2023). However, due to budget or

API constraints, models are typically evaluated on only a subset of the full range of available benchmarks. The selection is often made arbitrarily by the models' developers, making it challenging to compare models in a fair and holistic way (see Liang et al. 2023, Fig. 4). In response to this issue, we study the "small-bench" problem and hope it offers insights on efficient benchmarking of LLMs.

**Performance Prediction.** NLPERF (Xia et al., 2020) is a pilot work on performance prediction in NLP, focusing on bilingual and cross-lingual tasks. It demonstrates the potential of selecting an informative subset of tasks for evaluation, which inspired our work on searching for "small-bench." Ye et al. (2021) extend NLPERF to account for fine-grained performance measures, confidence intervals and calibration. Zhu et al. (2022) study predicting a model's downstream performance (GLUE, Wang et al. 2018) using probing tasks performance (SentEval, Conneau and Kiela 2018), and advocate for incorporating probing during pre-training. Our work aims to add to the discussion by focusing on the performance prediction of LLMs. Given the ongoing advancements and substantial influence of LLMs in the field of NLP, we believe this topic is both timely and relevant, potentially holding implications for future development of LLMs.

## 3 Performance Prediction on BIG-bench

### 3.1 Problem Definition

In this section, we focus on the problem of learning from experiment records of large language models, and predict the performance of an unseen combination of experiment settings.

**Notations.** We use $\mathcal{L}$ to denote the model families in consideration (*e.g.*, PaLM, GPT-3). We use $\mathcal{T}$ to denote the diverse collection of tasks in consideration (*e.g.*, 2-digit subtraction, emoji_movie). Formally, an experiment record is defined by the following values:

- Model family $l \in \mathcal{L}$
- Number of model parameters[2] $n_{param} \in \mathbb{N}$
- Task being evaluated on $t \in \mathcal{T}$
- Number of in-context examples $n_{shot} \in \mathbb{N}$

- Normalized performance[3] $y \in [0, 1]$

Our goal is to learn a regression model $f$ that predicts $\hat{y}$ based on $(l, n_{param}, t, n_{shot})$.

**Data Splits.** We obtain a large set of experiment records $D = \{(l, n_{param}, t, n_{shot}, y)\}$ and split them into three non-overlapping subsets, $D_{train}$, $D_{dev}$, $D_{test}$. By default, we use random splitting and adopt 10-fold cross validation.[4] In subsequent sections of this paper, we also use other splitting strategies for controlled analysis (§3.4 and §4).

**Evaluation.** We report root mean square error (RMSE) and coefficient of determination score ($R^2$) on the test set $D_{test}$. RMSE is defined as $\sqrt{\frac{1}{n}\sum_{i=1}^{n}(\hat{y}-y)^2}$. $R^2$ score characterizes the proportion of variance explained by the model, *i.e.*,

$$R^2 = \frac{\text{Explained Variance}}{\text{Total Variance}} = 1 - \frac{\sum_{i=1}^{n}(\hat{y}-y)^2}{\sum_{i=1}^{n}(\bar{y}-y)^2}$$
$$\text{where} \quad \bar{y} = \frac{1}{n}\sum_{i=1}^{n} y$$

In the main paper, we focus on RMSE and $R^2$ scores as they are widely accepted evaluation metrics for regression problems. In Appendix C.1 we introduce two alternative metrics based on rank correlation and discuss our findings.

### 3.2 Data

We construct our dataset $D$ from the BIG-bench repository.[5] We design a series of filtering criteria (*e.g.*, excluding tasks where *all* models have 0 accuracy, excluding tasks with <100 examples), which are detailed in Appendix A. After filtering, our dataset has 56,143 experiment records. We list high-level statistics about this dataset in Table 1.

We would like to highlight that this dataset covers *diverse* tasks and models. According to Srivastava et al. (2023), tasks in BIG-bench cover "problems from linguistics, childhood development, math, common-sense reasoning, biology, physics, social bias, software development, and beyond." We refer the readers to Fig. 3 and Table App. 3 in Srivastava et al. (2023) for an overview. We also made our best effort to incorporate all available model families in BIG-bench. The six model

---

[2]Here we use $n_{param}$ as the general representation of model scale. It is important to acknowledge a limitation: $n_{param}$ does not provide a comprehensive description of model scale. Pre-training compute and corpus size should be included should such information be available.

[3]For tasks metrics in a different range, *e.g.*, [0, 100], we normalize the score to be in [0, 1] for consistency.

[4]More specifically, we first split $D$ into 10 disjoint subsets, and then rotate on which ones are $D_{dev}$ and $D_{test}$.

[5]https://github.com/google/BIG-bench/

| | |
|---|---|
| # Experiment Records | 56,143 |
| # Model Families | 6 |
| | BIG-G T=0, BIG-G T=1, BIG-G Sparse, PaLM GPT-3, Gopher |
| # Models[†] | 51 |
| # BIG-bench Tasks | 134 |
| # BIG-bench Subtasks[‡] | 313 |
| $\{n_{shot}\}$ | $\{0, 1, 2, 3, 5\}$ |

Table 1: **Statistics of BIG-bench experiment records after filtering.** [†]"Model" is defined by model family $l$ and $n_{param}$, *e.g.*, PaLM 535B. [‡]The 313 subtasks fall into the 134 tasks. For simplicity we disregard the task-subtask hierarchy in this study. In the remaining of this paper, "tasks" refers to BIG-bench subtasks.

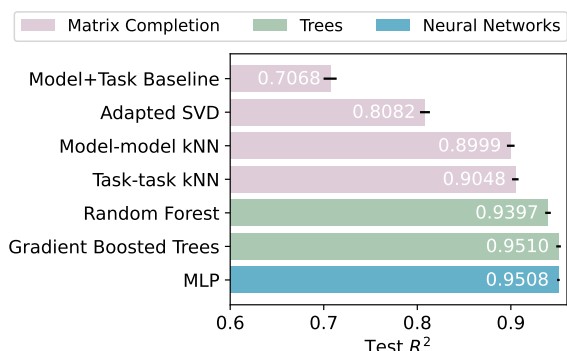

Figure 2: **Model Comparison for Performance Prediction using Default Data Split.** Gradient boosted trees and MLP achieve strong performance (RMSE $< 0.05$ and $R^2 > 0.95$).

families included are representative, and offer considerable diversity. We provide a brief summary of these model families (release date, company, model architecture choices, etc.) in Table 2.

### 3.3 Compared Models

**Matrix Completion.** Our problem is analogous to recommender systems or 2D matrix completion, if "models" (described by $l$ and $n_{param}$) are considered as "users," and "$n$-shot tasks" (described by $t$ and $n_{shot}$) are considered as "items." Each $n$-shot task (item $i$) is "rated" by each model (user $u$). With such adaptation, we consider the following matrix completion methods: **(a) Model + Task Baseline**: $\hat{y} = \mu + b_u + b_i$, where $\mu$ is the global average performance, $b_u$ represents the effect brought by user $u$, and $b_i$ represents the effect brought by item $i$. These parameters are learned by minimizing mean square error on $D_{train}$. **(b) Adapted SVD**: $\hat{y} = \mu + b_u + b_i + q_u^\top p_i$. Compared to (a), an additional vector $q_u$ is learned for each user $u$, and $p_i$ for each item $i$. The term $q_u^\top p_i$ is expected to model the interaction between user $u$ and item $i$. **(c) Model-model kNN**: First find the top $k$ models most similar to $u$, then aggregate the performance of these $k$ models on item $i$ by using weighted averaging. **(d) Task-task kNN**: similar to model-model kNN, but finds similar tasks and aggregate performance on these tasks instead.

**Trees.** We use two common tree-based methods that can directly learn to make predictions $\hat{y}$ from the input $(l, n_{param}, t, n_{shot})$: **(e) Random Forest** (Breiman, 2001) and **(f) Gradient Boosted Trees** (Friedman, 2001; Chen and Guestrin, 2016).

**Neural Networks.** We train simple **(g) multi-layer perceptron (MLP)** models to predict $\hat{y}$ from the input $(l, n_{param}, t, n_{shot})$. Hyperparameters such as number of layers and hidden dimensions are determined based on $D_{dev}$ performance.

**Featurization and Hyperparameters.** Featurization for trees and MLPs are explained in Appendix B. Hyperparameters and training details of these methods are discussed in Appendix D.1.

### 3.4 Results and Analysis

**Trees and MLPs achieve strong performance.** We experiment with the prediction models mentioned above and present their performance in Fig. 2. (1) Tree-based methods and MLP outperforms matrix completion methods by a large margin. We hypothesize that the 2D user-item simplification may cause loss of information on the input space. For example, the value of $n_{param}$ is merely used to distinguish different "users," and does not contribute to the computation of $\hat{y}$ directly. (2) Gradient boosted trees and MLP are the strongest among all compared models; both achieving RMSE $< 0.05$ (*i.e.*, on average mis-predict by $< 0.05$) and $R^2 > 0.95$ (*i.e.*, more than 95% of variance in $y$ is explained). This suggests that learnable patterns exist in LLM experiment records—LLM performance on unseen configurations can be predicted to a considerable extent in the current setting.

**Performance varies on different test groups (Fig. 3 ▇).** To have a more fine-grained understanding of the predictions, we group $D_{test}$ examples according to the features such as $n_{shot}$, $n_{param}$, and model family $l$, and then compute

$R^2$ on each of these test groups.[6] We use the MLP model predictions and present the results in Fig. 3 using dark blue bars (■). In terms of $n_{shot}$, we find that it is harder to predict zero-shot performance than 2- or 3-shot performance. In terms of the model family $l$, we believe the three BIG-G models (T=0, T=1, sparse) are easier to predict because their pre-training pipelines are similar. For $n_{param}$, we group all models into four buckets. For example, bucket 1 contains the smallest 25% models. We observe a trend that performance of larger models are harder to predict.

We also group $D_{test}$ according to whether the task $t$ is an emergent task (see Appendix E in Wei et al. 2022a). Our predictor achieves an $R^2$ score of 0.94 on the emergent group and 0.95 on the non-emergent group. This suggests in general emergent abilities are indeed harder to predict.

**Multi-group training is helpful (Fig. 3, / ↔ ■).** We further conduct a set of controlled experiments by only training on examples from the test group of interest (*e.g.*, examples with $n_{shot} = 0$). We name them as *single-group* experiments ( / ), as opposed to *multi-group* experiments (■) done in previous sections where the predictor is trained on all groups. Notably, in all settings, multi-group $R^2$ is always larger than single-group $R^2$. There are limited observations of Gopher models in the training set, and they benefit from multi-group learning significantly ($R^2$ increases from 0.74 to 0.87). This reaffirms the claim that LLM performance exhibits shared patterns across various settings.

**Some groups benefit more from multi-group training, some are intrinsically harder to predict. (Fig. 3, ■ ↔ / ↔ ■)** Our controlled experiments also allow us to distinguish between two factors: the group is intrinsically harder to predict *or* the group benefits more from multi-group learning. In Fig. 3(a), results suggest that $n_{shot} = 0$ group is not necessarily harder to predict than $n_{shot} = 2$ and $n_{shot} = 3$ on its own (indicated by / bars), but the latter two benefit more from multi-group learning. Typically, when evaluating LLMs on a task, there is a huge performance boost when going from zero-shot to one-shot, and the performance improves more stably when more shots become available. It is easier to predict 3-shot performance when given 0,1,2-shot performance, than to predict

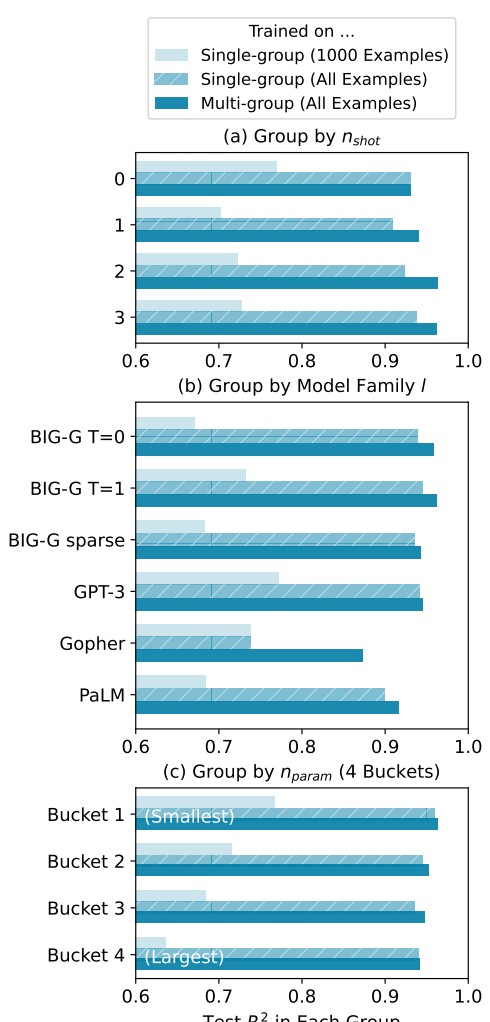

Figure 3: $R^2$ **Score when Grouped with** $n_{shot}$, $l$, **and** $n_{param}$. **Example:** ■ Multi-group $n_{shot} = 0$ means training on the complete $D_{train}$ (containing all $n_{shot}$ values) and evaluate on $n_{shot} = 0$ examples in $D_{test}$. ■ Single-group (1000 Examples) $n_{shot} = 0$ means using 1000 $n_{shot} = 0$ examples in $D_{train}$ as the training data.

0-shot performance when given 1,2,3-shot performance. This partly explains why the 0-shot group does not benefit much from multi-group training.

In Fig. 3(b), BIG-G T=0 and BIG-G T=1 benefit from multi-group learning more than the GPT-3 model family, resulting in higher $R^2$ scores in the multi-group setting. In Fig. 3(c), we observe that larger models tend to be intrinsically more challenging to predict. This observation is more significant when the single-group training set size is controlled to be 1000 (represented by ■ bars), where we observe a clear trend that groups consisting of larger models achieve lower $R^2$ score.

**Identifying most/least "predictable" tasks.** We further experiment with grouping test examples ac-

---

[6]Note that the denominator in $R^2$, total variance, will be different for each group.

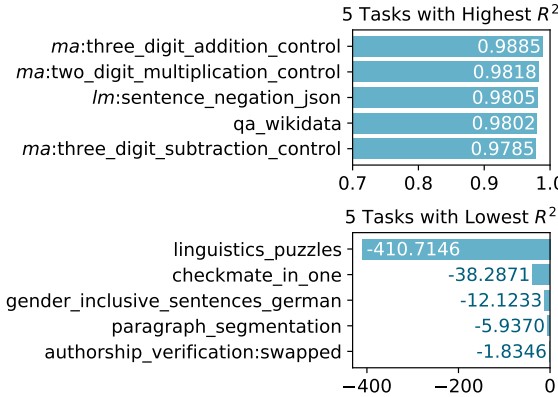

Figure 4: **5 Most and Least "Predictable" Tasks Based on $R^2$ Score.** *ma* stands for "modified_arithmetic", *lm* stands for "linguistic_mapping". Fig. 9 and 10 illustrate their scaling behaviors.

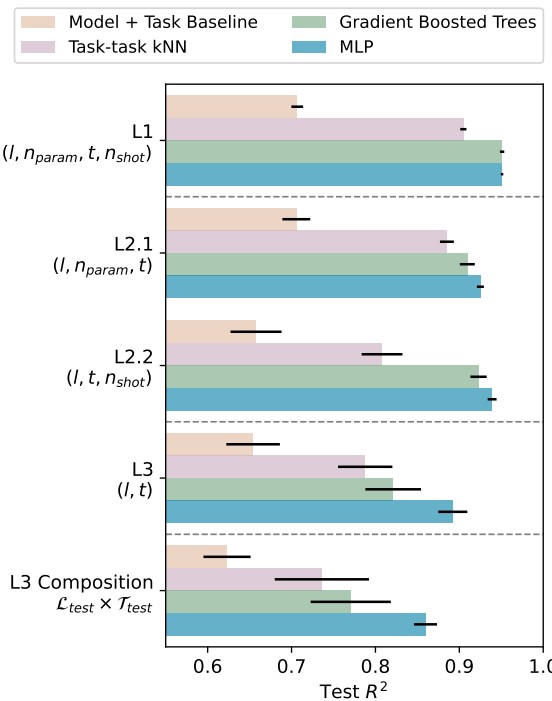

Figure 5: **Performance of Different Prediction Models on Challenging Train-Test Splits.** As the setting becomes more challenging (L1 → L2 → L3 → L3 Composition), performance gradually drops and variance increases. MLP is least sensitive to these changes.

cording to the task $t$ they belong to, to identify the most and least predictable tasks. The five most predictable tasks (Fig. 4 Top) include qa_wikidata and linguistic_mappings, which were tasks marked as having high *linearity*[7] in their scaling behavior (Srivastava et al. 2023, Sec 3.4). This observation is reasonable because high linearity typically implies predictability. However, modified_arithmetic, a task marked as having high *breakthroughness* (Srivastava et al., 2023) and *emergence* (Wei et al., 2022a), is considered highly predictable in our setting. Our hypothesis is that the predictor is able to infer this by learning from experiment records with similar configurations: If breakthroughness is observed with other models or other number of shots, the trained predictor is able to infer the trend for a new experiment configuration. We believe it is still challenging to predict breakthroughness/emergence in more restricted setting, *e.g.*, based on task meta-data or input text alone.

For the five tasks with lowest $R^2$ scores (Fig. 4 Bottom), we manually examined their scaling behavior (Fig. 10) and indeed find these curves to be surprising. For future work, it will be interesting to investigate the underlying reasons, and to identify common characteristics shared among these tasks.

## 4 Creating Challenging Train-Test Splits

Previously, we randomly split $D$ into $D_{train}$, $D_{dev}$, and $D_{test}$, *i.e.*, we randomly sampled

---

[7]Linearity measures "the extent to which performance improves reliably with scale;" Breakthroughness measures "the extent to which a model is able to learn a task only once the model grows beyond a critical scale." See Srivastava et al. (2023) Appendix B for the formal definition.

$(l, n_{param}, t, n_{shot})$ combinations. This is a relatively easy setting; for example, when the model family $l$, number of parameters $n_{param}$, and task $t$ are kept the same, it may be easy for a model to predict performance of $n_{shot} = 2$ when the records of $n_{shot} = 1$ and $n_{shot} = 3$ appear in $D_{train}$. A more challenging data split would ensure that the combinations of $(l, n_{param}, t)$ in the test set are completely unseen in $D_{train}$, and the model is required to predict for all possible $n_{shot}$ values. Taking it a step further, one may want make predictions on an unseen configuration of $(l, t)$ for all possible values of $n_{param}$ and $n_{shot}$.

### 4.1 Train-Test Split Settings

To simulate these use cases, we design and compare model performance on three additional settings (L2.1, L2.2, L3).
- L1: Random $(l, n_{param}, t, n_{shot})$, used in §3
- L2.1: Group by $(l, n_{param}, t)$
- L2.2: Group by $(l, t, n_{shot})$
- L3: Group by $(l, t)$

For example, in L3, we first group all experiment records in $D$ according to $(l, t)$, then create $D_{train}$, $D_{dev}$, and $D_{test}$ by random splitting the groups.

Additionally, we make L3 setting even more challenging by holding out one entire subset of $\mathcal{L}_{test} \times \mathcal{T}_{test}$. Specifically, we first select $\mathcal{L}_{test}$, a subset of model families $\subseteq \mathcal{L}$, and $\mathcal{T}_{test}$, a subset of tasks $\subseteq \mathcal{T}$. After this, $D_{test}$ is defined as $\{(l, n_{param}, t, n_{shot}, y)|l \in \mathcal{L}_{test}, t \in \mathcal{T}_{test}\}$. This corresponds to a practical scenario where a new model family is developed, and researchers want to take a "sneak peek" at the full picture of its capabilities—evaluate on a subset of tasks (i.e., $\mathcal{T}_{train} = \mathcal{T} \backslash \mathcal{T}_{test}$), and predict the model's performance on the remaining tasks (*i.e.*, $\mathcal{T}_{test}$). We refer to this as the "L3 Composition" setting.[8]

## 4.2 Results and Analysis

**Main Results.** Results on four representative models in these settings are visualized in Fig. 5. We observe that as the settings becomes more challenging (L1 $\rightarrow$ L2 $\rightarrow$ L3 $\rightarrow$ L3 Composition), performance gradually decreases and the standard deviation increases. Another important observation is that though MLP and gradient boosted trees are comparable in the L1 setting, MLPs are less sensitive to the increased difficulty (performance decrease is smaller and standard deviation is smaller).

**Sample Prediction Results in L3.** In Fig. 7 we visualize predictions on four sample $(l, t)$ combinations by the MLP model—two achieving high $R^2$ scores and two achieving low $R^2$ scores. We have two high-level observations: (1) Predictions are more accurate on $(l, t)$ combinations which has observations on similar tasks $t'$ or similar models families $l'$ in $D_{train}$. (2) Over-estimation is a common type of mistake made by our trained prediction models. We observe several cases of "false positives" of emergent abilities. Due to space limit, we defer more discussion in §C.2.

## 5 Searching for "small-bench"

There has been a recent emphasis on assessing the *generality* of large language models, which involves evaluating these models on numerous tasks and scenarios. However, it will be extremely expensive to conduct all these experiments every time a new model is developed in the future. Extending from the holding out $\mathcal{L}_{test} \times \mathcal{T}_{test}$ setting in §4.1, in this section, we formulate and study the problem of searching for "small-bench:" Can we find a subset of BIG-bench tasks, such that when

a new model family is evaluated on it, the performance of the remaining tasks can be maximally recovered? In the following we give a formal definition of this problem (§5.1), construct "small-bench" candidates using different search algorithms and strategies (§5.2), and present our findings (§5.3).

## 5.1 Problem Definition

Our goal is to find $\mathcal{T}_{train}$, a subset of all tasks $\mathcal{T}$, that are selected and used for evaluating new model families $\mathcal{L}_{test}$. We use $b$ to represent the evaluation budget, *i.e.*, $|\mathcal{T}_{train}| = b$. We use $\mathcal{T}_{test} = \mathcal{T} \backslash \mathcal{T}_{train}$ to denote the tasks whose performance we wish to recover. The problem of finding the optimal $T_{train}^{(b)*}$ can be formulated as the following:

$$\underset{\mathcal{T}_{train}}{\arg\max} \quad R^2(\mathcal{T}_{test} \times \mathcal{L}_{test})$$
$$\text{s.t.} \quad \mathcal{T}_{train} \subseteq \mathcal{T}, \quad |\mathcal{T}_{train}| = b$$

$R^2(\mathcal{T}_{test} \times \mathcal{L}_{test})$ represents the $R^2$ score on $\mathcal{T}_{test} \times \mathcal{L}_{test}$ when a predictor is trained on the remaining experiment records, as previously done in §4.

**Evaluation.** Ideally the optimal $\mathcal{T}_{train}$ should allow us to predict the performance of *any* new model family, without overfitting to a specific held-out model family. To evaluate this, we adopt nested cross-validation on $\mathcal{L}$ during evaluation of a selected $\mathcal{T}_{train}$. Specifically, given that $|\mathcal{L}| = 6$, we create $6 \times 5 = 30$ different ways to hold out one model family as $\mathcal{L}_{dev}$ and one model family as $\mathcal{L}_{test}$. We then train 30 prediction models and report the average of 30 $R^2(\mathcal{T}_{test} \times \mathcal{L}_{test})$ scores.

## 5.2 Compared Methods

We consider different evaluation budget $b \in \{4, 8, 16, 24, 32, 42\}$. We compare the following methods for selecting $\mathcal{T}_{train}$.

**Baselines.** **(a) BIG-bench Lite** (Srivastava et al., 2023): A subset of BIG-bench for cheaper evaluation, proposed in the original BIG-bench paper. $|\mathcal{T}_{train}| = 42$ for BIG-bench Lite.[9] **(b) BIG-bench Hard** (Suzgun et al., 2023): A subset of BIG-bench containing challenging tasks that cannot be solved with direct in-context learning but can be improved with chain-of-thought prompting (Wei et al., 2022b). $|\mathcal{T}_{train}| = 24$ for BIG-bench Hard. **(c) Random**: For each $b$, randomly sample 5 $\mathcal{T}_{train}$

---

[8]In §D.2, we describe our efforts to ensure that the performance is as comparable as possible across settings.

[9]Tasks in BIG-bench Lite/Hard are listed in Appendix G. To the best of our knowledge, the selection process for BIG-bench Lite is not disclosed.

such that $|\mathcal{T}_{train}| = b$. We report the mean and standard deviation of these 5 runs.

**Search Algorithms.** **(d) Greedy Search**: Based on the search result $\mathcal{T}_{train}^{(b-1)}$ at budget $b-1$, enumerate all tasks not present in $\mathcal{T}_{train}^{(b-1)}$, and select the one task that achieves the highest $R^2(\mathcal{T}_{test}^{(b)} \times \mathcal{L}_{test})$ to form the $\mathcal{T}_{train}^{(b)}$ at budget $b$. **(e) Best of 5000**: For each $b$, randomly select 5000 $\mathcal{T}_{train}$ and select the one achieving the highest $R^2(\mathcal{L}_{dev} \times \mathcal{T}_{test})$.

Note that these search algorithms optimize $R^2(\mathcal{T}_{test} \times \mathcal{L}_{dev})$ during search, to ensure that $\mathcal{T}_{test} \times \mathcal{L}_{test}$ is held-out for evaluation. Additionally, to make the search computationally tractable, we only use 1 fixed fold from the 30 folds during search. We discuss the impact of these experimental decisions in Appendix C.5.

**Clustering-based.** We hypothesize that a good "small-bench" should be *diverse* (covering the task space comprehensively while avoiding redundancy by excluding similar tasks) and *representative* (each selected task providing informative insights for recovering the performance of other tasks). To validate this, we use the following methods to construct $T_{train}$. **(f) $k$-means:** We extract the task representations[10] learned by our MLP models in §3. We apply $k$-means clustering to these representations, group them into $b$ clusters, and then select the task closest to the centroid of the each cluster. **(g) $k$-means + Task Value:** We first calculate the task value for each task in $\mathcal{T}$ by aggregating their contributions from the Best of 5000 search history. For example, if a task is present in 20 trials out of the 5000, its task value will be the average of the $R^2$ scores from those 20 trials. We then incorporate this information into $k$-means clustering, by selecting the task closest to the centroid among tasks that are top 25% valuable globally.

### 5.3 Results and Discussion

We visualize the results of all compared methods in Fig. 6. We have the following key observations.

**BIG-bench Hard and Lite are sub-optimal for recovering performance on remaining tasks.** A 8-task subset found by Best of 5000 and randomly-sampled 16-task subsets can match the 24-task BIG-bench Hard for this goal. We further examine the re-

---

[10]The task $t$ is represented as an one-hot feature in the input space. Thus, for each task, there is a vector corresponding to each task in the first layer of the MLP. We refer to these as "task representations."

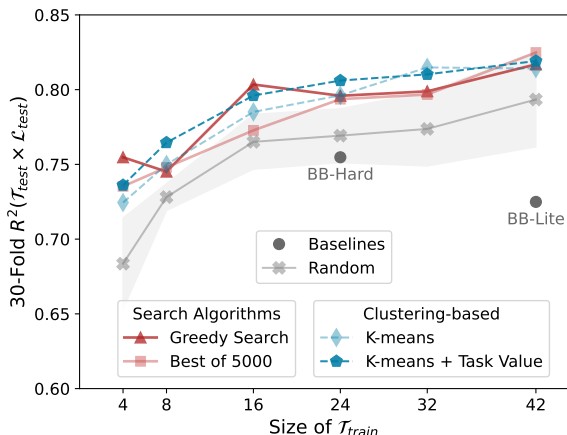

Figure 6: **"Small-bench" Search Results.** X-axis: size of "small-bench" ($\mathcal{T}_{train}$), *i.e.*, number of tasks selected for evaluating a new model family. Y-axis: $R^2$ score on recovering performance of remaining tasks. The complete BIG-bench will be at $(313, 1.0)$ in this figure. **Takeaways:** (1) BIG-bench Lite and Hard are sub-optimal for recovering performance on remaining tasks; (2) Task diversity and task value are important for constructing effective "small-bench" candidates.

sults and find several cases where BIG-bench Hard fails to represent the complete BIG-bench. For example, according to full BIG-bench performance recovered from BIG-bench Hard, BIG-G T=1 2B is better than GPT-3 Large; however, according to the ground-truth BIG-bench performance, GPT-3 Large is better than BIG-G T=1 2B, which is captured more accurately when a 24-task small-bench candidate is used. See Table 3 for the details.

It is important to note that BIG-bench Hard was *not* specifically designed for our goal, and thus is not expected to be competitive in our problem. Yet it is surprising that it underperforms randomly-sampled subsets. As a general recommendation for evaluating newly-developed models, we suggest using the $\mathcal{T}_{train}$ subsets found by solving the optimization problem in §5.1. If there is a specific evaluation goal in mind (*e.g.*, focus on frontier, as in the case of BIG-bench Hard), $\mathcal{T}_{train}$ should still be manually selected.

**Greedy search is unstable and finds sub-optimal solutions.** Search algorithms consistently outperforms randomly sampled $\mathcal{T}_{train}$ sets; however, greedy search appears to be unstable, with occasional performance drops as the budget increases. Furthermore, at $b = 42$, it underperforms the Best of 5000 approach. We include additional results on other search algorithms, including beam search

and simulated annealing, in §C.4, where we observe similar instablitiy. One possible explanation is the complexity of the search space, where the greedy search algorithm cannot guarantee finding the optimal solution. The gaps between the search objective ($\mathcal{L}_{dev} \times \mathcal{T}_{test}$ in one fold) and the evaluation objective ($\mathcal{L}_{test} \times \mathcal{T}_{test}$ in 30 folds) could also contribute to this issue (§C.5).

**Task diversity and task value are important factors for constructing "small-bench."** Firstly, $k$-means is comparable to or surpasses Best of 5000, despite that it is not explicitly optimized for the $R^2$ objective. This supports the notion that diversity is an important factor in constructing "small-bench." This finding also suggests that the MLP models for performance prediction produce meaningful task representations as a side product. Secondly, $k$-means + Task Value is comparable to or outperforms $k$-means, confirming that task value is another important factor for constructing "small-bench," complementing the diversity aspect.

## 6 Conclusion and Future Work

In this work, we began with the question, "How predictable are large language model capabilities?" and conducted a detailed case study on BIG-bench. We first formulated the machine learning problem of predicting performance given configurations such as model family, the number of parameters, task, and the number of in-context learning examples. Our strongest prediction model achieves an $R^2$ score greater than 95%, which suggests past LLM experiment observations can be used to predict the performance of new experiment configurations. To address the problem of increasing evaluation cost on massively multi-task benchmarks, we introduced the problem of searching for an informative "small-bench." Results suggest that popular subsets such as BIG-bench Lite and BIG-bench Hard are not optimal for this purpose. Instead, subsets characterized by diversity and high task values offer competitive "small-bench" candidates, highlighting the importance of these two factors.

In closing, while our study primarily focused on the predictability of LLM capabilities, we hope to initiate discussions on the following broader topics.

**Rethinking LLM Evaluation.** Currently, there is a lack of consensus regarding evaluation practices for newly developed LLMs. Often times new LLMs are evaluated on different set of selected tasks, making it hard to compare different models and quantify the progress in LLM development. Moreover, task selection is often heuristic, following past practices, or chosen arbitrarily without principled justifications. We anticipate more active discussion on establishing evaluation practices that assess LLM capabilities efficiently, reliably and rigorously, and we hope our work provides useful insights towards this. Related to our efforts on searching for "small-bench," Perlitz et al. (2023) investigate the impact of benchmarking options on the trade-off between computational efficiency and reliability, and develop Flash-HELM, an efficient alternative to HELM (Liang et al., 2023). Vivek et al. (2023) propose Anchor Point Selection to select representative examples in the test set and reduce evaluation cost at the instance-level.

**Broadening observations on LLM capability landscape.** Complementary to BIG-bench, several ongoing initiatives, such as HELM (Liang et al., 2023), Open LLM Leaderboard[11], and Eleuther AI LM Harness[12] are dedicated to systematically evaluating existing LLMs. Integrating insights from these great initiatives into future work has the potential to enhance the accuracy of LLM performance prediction and deepen our understanding of LLM capabilities. Additionally, it would be intriguing to take into account recent advances such as chain-of-thought prompting (Wei et al., 2022b) and instruction tuning (Sanh et al., 2022; Ouyang et al., 2022), and systematically measure their effects on LLM capabilities.

## Limitations

**Limited to BIG-bench results.** We choose BIG-bench for our study due to its extensive collection of experiment records. Though it offers considerable diversity in terms of tasks and models, several limitations exist. (1) Tasks: It's important to note that BIG-bench tasks are sourced from the research community and may not accurately reflect the actual distribution of tasks encountered by LLMs in real-world scenarios. Therefore, our study has limitations in terms of generalizing our conclusions to the real-world task distribution. (2) Models: Though we have made every effort to incorporate as many model families as possible, there are only

---

[11]https://huggingface.co/spaces/HuggingFaceH4/open_llm_leaderboard

[12]https://github.com/EleutherAI/lm-evaluation-harness

6 model families in our experiment record dataset derived from BIG-bench. Such scarcity introduces instability and increases the difficulty in investigating the "small-bench" problem.

**Limited to publicly-available LLM meta-data.** LLMs capabilities are dependent on many factors, beyond the model family $l$ and number of parameters $n_{param}$ used in this study. Factors such as pre-training stability, convergence, pre-train corpus composition, etc., all play important roles. However, we often don't have access to this information. In this work, we assume that the input features $(l, n_{param})$ can capture such information during training implicitly. In the future, we believe our method can be expanded to include additional pre-training meta-data when they become available.

**Limited to interpolation settings.** Our experiments mainly concentrate on interpolation settings, where the combinations of $(l, n_{param}, t, n_{shot})$ are new in the test set, but each of the input element is seen at least once in the training set. As LLMs continue to grow in size, a very important aspect is predicting performance of larger models in an extrapolation setting. We present some preliminary findings in this setting in §C.3.

**Limitations in evaluation metrics.** We use RMSE and $R^2$ as they are widely-used metrics for regression tasks. However, both metrics have their limitations for our problem, especially in the context of conducting group-wise comparison (§3.4). RMSE does not account for the variance in the target variable. A low RMSE value for a task may be solely due to the fact that the task performance is relatively insensitve to different experiment settings. On the other hand, while $R^2$ score accounts for variance, it creates discrepancies when conducting group-wise comparison since the denominator used to compute $R^2$ differs for each group. To get a more comprehensive picture of our prediction model, we introduce task-average Pearson Correlation and Kendall Rank Correlation for evaluation and discuss our findings in Appendix C.1.

## Acknowledgements

We thank anonymous reviewers and members of USC NLP for their valuable feedback. QY and XR were supported in part by the Office of the Director of National Intelligence (ODNI), Intelligence Advanced Research Projects Activity (IARPA), via the HIATUS Program contract #2022-22072200006, the DARPA MCS program under Contract No. N660011924033, the Defense Advanced Research Projects Agency with award W911NF-19-20271, NSF IIS 2048211, and gift awards from Google and Amazon. HF was supported by a USC Provost Fellowship Award. RJ was supported by an Open Philanthropy research grant and a Cisco Research Award.

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

## A  Filtering BIG-bench Records

We use the following criteria when filtering the records. After filtering we obtain a dataset with 56k+ records which is described in Table 1. For now we limit the scope to predicting performance on the preferred evaluation metric.

1. Keep json tasks and remove programmatic tasks.

2. Remove T0/T5 models because there are only 14 records for each of these two. Keep BIG-G, PaLM, GPT-3, Gopher models.

3. Remove BIG-bench subtasks whose performance on the preferred metric is zero for all models.

4. Keep experiments whose preferred metric is in `exact_str_match`, `multiple_choice_grade`, `rougeLsum`. This keeps 93% of all records before this step.

5. Remove entries of aggregating results from multiple *subtasks* as the performance of a *task*.

6. Remove subtasks with less than 100 examples because small sample size may lead to large variance during evaluation

We present a summary of the 6 model families in these records in Table 2.

## B  Featurization

In the main body of the paper we use the abstraction of $(l, n_{param}, t, n_{shot})$ to describe experiment configurations. In the actual training of tree-based models and MLP, we modify how these features are represented. In particular, the input features contain the following:

1. $l$ is converted as binary features for each model family, *e.g.*, `is_PaLM`.

2. $(l, n_{param})$ is converted as binary features for each model, e.g., `is_PaLM_535B`.

3. There are 6 numerical features for the number of parameters: number of total parameters, number of non-embedding parameters, number of FLOP-matched non-embedding parameters; and the natural log of these three values.

4. $t$ is converted as binary features for each task, *e.g.*, `is_code_line_description`.

5. $m$ is a binary feature for the preferred metric associated with the task $t$, *e.g.*, `is_exact_str_match`. BIG-bench defines a preferred metric for each task, so in the abstraction it is covered by $t$.

6. $n_{shot}$ is directly used as an input feature.

For numerical features (6 features for the number of parameters and 1 feature for the number of shots), we use `StandardScaler` in the sklearn libarary to normalize them. Additionally, we normalize the performance value $y$ to

be in the range of $[0, 1]$. `exact_str_match` and `multiple_choice_grade` already satisfy this constraint. The reported `rougeLsum` values are in the range of $[0, 100]$ and are multiplied by 0.01 to form our dataset.

## C  Extended Experiment Details and Results

### C.1  Additional Evaluation Metrics for Performance Prediction

In addition to RMSE and $R^2$ score, two common metrics for evaluating regression models, we introduce two new metrics, Task-average Pearson Correlation and Task-average Kendall Rank Correlation. The usage of Pearson Correlation and Kendall Rank Correlation are inspired by Liu et al. (2023) and we further adapt them to be averaging across tasks.

Concretely, we first group the test set $D_{test}$ into $|\mathcal{T}|$ groups, based on the associated task $t$ of each example. Within each group, we compute the Pearson Correlation or Kendall Rank Correlation between the predicted performance and the actual performance. Finally, the average of these rank correlation values across all tasks $\mathcal{T}$ is then reported as the task-average rank correlation. We report these numbers along with RMSE and $R^2$ in Table 6.

Generally, prediction models with higher $R^2$ scores exhibit higher rank correlation. Exceptions emerge when closely comparing tree-based models and MLP models. While MLP models are comparable or outperform tree-based models in terms of $R^2$ score, tree-based models tend to outperform MLP models in terms of task-avg (rank) correlation. Our further investigation reveals that tree-based models make more errors with large absolute errors, which are penalized heavily by RMSE and $R^2$ (involving taking square of these errors), whereas rank correlation is less sensitive to such errors.

Throughout our paper, we primarily focus on experimentation with MLP models due to their faster runtime and the values of the learned representations in MLP. In practice, we recommend selecting methods based on the final goal: MLP models for more accurate prediction in terms of *exact values*; tree-based methods for more accurate *ranking* of different experiment settings.

### C.2  Sample Predictions in the L3 setting

In Fig. 7 we visualize predictions on four sample $(l, t)$ combinations by the MLP model. The left

| Model Family | Company | Release Date | Pre-train Corpus | # Parameters | Architecture Choices | Infrastructure |
|---|---|---|---|---|---|---|
| BIG-G (T=0, T=1) | Google | Jun 2022 | 2.8T tokens | 2M to 137B | Gated-GELU, relative attention | TPUv3, GSPMD |
| BIG-G Sparse | Google | Jun 2022 | 2.8T tokens | 51M to 46B | BIG-G + Mixture-of-Experts | TPUv3, GSPMD |
| PaLM | Google | Apr 2022 | 780B tokens | 8B, 62B, 540B | SwiGLU activation, parallel layers, etc. | TPUv4, Pathway |
| GPT-3 | OpenAI | May 2020 | 300B tokens | 125M to 175B | Alternating dense and sparse attention | GPUs |
| Gopher | DeepMind | Nov 2021 | 300B tokens | 44M to 280B | RMSNorm, relative position encoding | TPUv3, model parallelism, etc. |

Table 2: **Summary of model families included in this study.** These model families offer considerable diversity. Information in this table is aggregated from Srivastava et al. (2023); Chowdhery et al. (2022); Brown et al. (2020); Rae et al. (2021)

| | BIG-G T=1 2B Wins | Tie | GPT-3 Large Wins | Conclusion |
|---|---|---|---|---|
| Performance Recovered from BBH (24 subtasks) | 20.1% | 70.1% | 9.8% | BIG-G T=1 2B is better |
| Performance Recovered from small-bench (24 subtasks) | 19.1% | 56.3% | 24.6% | GPT-3 Large is better |
| Ground-truth full BIG-bench Performance | 14.3% | 55.3% | 30.4% | GPT-3 Large is better |

Table 3: **Using BIG-bench Hard and "small-bench" to recover performance and compare models.** In this example, BBH is less informative in recovering performance on remaining tasks, and thus is the comparison is less accurate.

two are cases achieving high $R^2$ scores. The right two are cases achieving low $R^2$ scores.

For $(l, t)$ combinations achieving high $R^2$ scores, our observation is that either a combination $(l, t')$ exists in $D_{train}$ such that $t$ and $t'$ are similar (e.g., $t$ and $t'$ are two sub-tasks from the same BIG-bench task), or $(l', t)$ exists in $D_{train}$ such that $l'$ and $l$ are similar (e.g., both $l$ and $l'$ from the three BIG-G model families). Our interpretation is that the learned predictor is capturing model family similarities and task similarities and therefore predicts more accurately when $l$ or $t$ has a similar counterpart in the training set.

For $(l, t)$ combinations achieving low $R^2$ scores, we observe several cases of overestimating performance or predicting "false positives" of emergent abilities. The selection of these combinations are largely due to using $R^2$ as selection criteria—they have small total variance as the denominator for $R^2$ score, so any overestimation will results in an extremely negative $R^2$ score. Nevertheless, these qualitative results suggest that overestimation is a common type of mistake made by our trained prediction model.

## C.3 Performance Prediction in Extrapolation Settings

In the input space of our problem, $n_{shot}$ and $n_{param}$ are numerical features. Thus it is possible to test the extrapolation capabilities on these two inputs. We create three settings for testing the model's extrapolation capabilities:

- E1: $D_{test}$ contains examples with $n_{shot} = 3$, $D_{train}$ contains examples with $n_{shot} = 0, 1, 2$
- E2.1: $D_{test}$ contains examples with

- $(l, n_{param}) = (\text{GPT-3}, 200B)$
- E2.2: $D_{test}$ contains examples with $(l, n_{param}) = (\text{PaLM}, 535B)$

**Relaxing Constraints by leaking 10% $D_{test}$.** Pure extrapolation may be extremely challenging. To better contextualize the results and understand limitations, we compare model performance in three slightly different settings. The first setting (S1) is holding 10% $D_{train}$ as dev set for hyperparameter selection and early stopping. This corresponds to the pure extrapolation setting, as no information about $D_{test}$ is available at training time. The second setting (S2) is holding 10% $D_{test}$ as dev set. Information about $D_{test}$ is indirectly leaked to model training. The third setting is leaking 10% $D_{test}$ during training. This third setting (S3) is mainly for reference.

More specifically, we split the original $D_{train}$ into 90% $D'_{train}$ and 10% $D_{dev1}$. Also we split the original $D_{test}$ into 90% $D'_{test}$ and 10% $D_{dev2}$. In S1, model training, selection and evaluation is done with $(D'_{train}, D_{dev1}, D'_{test})$. In S2, we allow model selection with $D_{dev2}$, so the experiment is done with $(D'_{train}, D_{dev2}, D'_{test})$. In S3, we leak $D_{dev2}$ (which is from the test distribution) to training time, and the experiment is done with $(D'_{train} \cup D_{dev2}, D_{dev1}, D'_{test})$.

**Results and Findings.** We present the results in Table 4. (1) Extrapolation in terms of $n_{shot}$ is promising, achieving $R^2$ which is greater than 0.9. However, extrapolation to increased model size remains challenging. This is closely related to the observation that emergent abilities is difficult to predict (Ganguli et al., 2022; Wei et al., 2022a).

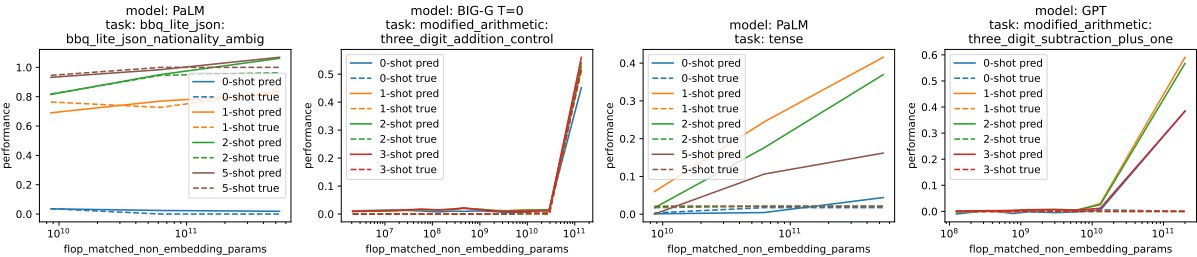

Figure 7: **Sample Prediction Results in L3 setting (§C.2). Left 2**: $(l, t)$ combinations achieving high $R^2$ score; **Right 2**: $(l, t)$ combinations achieving low $R^2$ score.

(2) Performance in S2 is consistently better than S1. Note that the only difference between these two setting is the $D_{dev}$ used to do model selection. This suggests that the model overfits and fails to extrapolate well in the strict S1 setting. Leaking some information about the test distribution (as in S2 and S3) can greatly help improve prediction accuracy.

## C.4 Additional Results on "small-bench" search

We experiment with two additional search algorithms for "small-bench" search: **(1) Randomized Beam Search**: Similar to regular beam search, except that we maintain a beam size of $q = 4$ and we enumerate $1/q$ randomly selected task candidates at each timestamp. This ensures the search runtime is equivalent to greedy search. **(2) Simulated Annealing** (Černỳ, 1985): Initialize with a seed $T_{train}$; at time $t$, iteratively search in the neighbourhood of the $T_{train}$ at time $t - 1$ and occasionally allowing uphill moves (*i.e.*, moving towards a worse solution). Results are visualized in Fig. 8. Similar to greedy search, these two search methods face optimization challenges discussed in §5.3 and may lead to sub-optimal solutions.

## C.5 Gaps between the "small-bench" search objective and evaluation objective

In §5.3, we observe that the performance of greedy search is unstable and hypothesize that this is partly due to the gaps between the search and evaluation objective.

**Gap between $\mathcal{T}_{dev}$ and $\mathcal{T}_{test}$.** To simulate the scenario that the prediction model is expected to make predictions on an unseen model family, we make sure that the search algorithm optimizes on $\mathcal{T}_{dev} \times \mathcal{L}_{test}$, and holds out $\mathcal{T}_{test} \times \mathcal{L}_{test}$ for evaluation. This creates a dev-test shift which may affect search algorithm results.

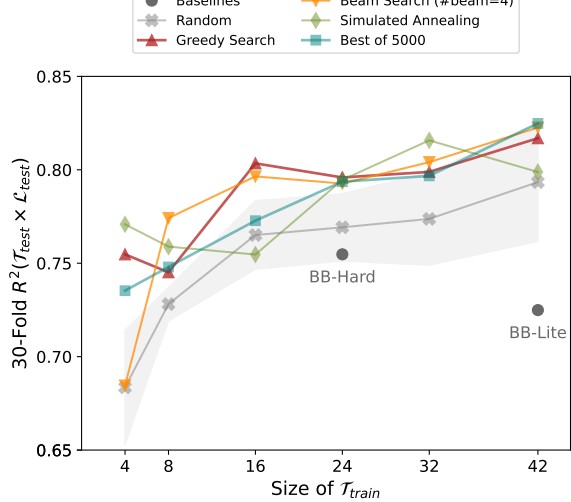

Figure 8: **Additional "Small-bench" Search Results.** See §C.4 for in-depth analysis.

**Gap between 1 fold and 30 folds.** Due to runtime concerns, we only launch search algorithms on 1 fold, while at evaluation time we use all 30 folds. The 30 folds, derived from 6 distinct model families, may exhibit significant variations. Consequently, the search result on 1 fold may overfit to that specific fold and become less optimal on all 30 folds.

## D Reproducibility

### D.1 Hyperparameters and Training Details

For all the following methods and for each data split setting (L1/L2.1/L2.2/L3/L3 Composition), we select hyperparameters based on the dev performance on the first fold, and select the best set of hyperparameters from 100 random combinations from a pre-defined list.

For Random Forest, we use the implementation of RandomForestRegressor from sklearn library. We use squared_error as optimization criterion. The hyperparameter candidates are sampled

| | S1: $D_{dev}$ is 10% $D_{train}$ | | S2: $D_{dev}$ is 10% $D_{test}$ | | S3: Leak 10% $D_{test}$ | |
|---|---|---|---|---|---|---|
| | RMSE ($\downarrow$) | $R^2$ ($\uparrow$) | RMSE ($\downarrow$) | $R^2$ ($\uparrow$) | RMSE ($\downarrow$) | $R^2$ ($\uparrow$) |
| **More shots** | | | | | | |
| E1: Hold out $n_{shot} = 3$ | 0.0618 | 0.9297 | 0.0587 | 0.9367 | 0.0545 | 0.9454 |
| **Scaling up model size** | | | | | | |
| E2.1: Hold out GPT-3 200B | 0.1314 | 0.6968 | 0.1017 | 0.8187 | 0.0830 | 0.8791 |
| E2.2: Hold out PaLM 535B | 0.2708 | 0.1535 | 0.1923 | 0.5731 | 0.1668 | 0.6789 |

Table 4: Results of Extrapolation Experiments (§C.3).

from the following:

```
1  {
2      "n_estimators": [30, 100, 300],
3      "max_depth": [None, 16, 32, 64, 128],
4      "min_samples_split": [2, 4, 8],
5      "min_samples_leaf": [1, 2, 4],
6      "max_features": [1.0, 0.9, 0.8, 0.7, 0.6, "
           sqrt"],
7      "max_samples": [1.0, 0.9, 0.8, 0.7, 0.6]
8  }
```

For Gradient Boosted Trees, we use the implementation of `XGBRegressor` from `XGBoost` library (Chen and Guestrin, 2016). We use the `reg:squarederror` as objective. The hyperparameter candidates are sampled from the following:

```
1  {
2      "n_estimators": [30, 100, 300, 1000],
3      "learning_rate": [0.1, 0.3, 0.5, 0.8, 1.0],
4      "max_depth": [None, 16, 32, 64, 128], # None
           indicates no limit
5      "gamma": [0, 0.1, 0.2],
6      "subsample": [0.6, 0.7, 0.8, 0.9, 1.0],
7  }
```

For MLP, we implement with Pytorch. We use Adam optimzier and use MSELoss. The hyperparameter candidates are sampled from the following:

```
1   {
2       "lr": [1e-3, 3e-4, 1e-4],
3       "batch_size": [32, 64, 128],
4       "dropout": [0.0, 0.05, 0.1, 0.15, 0.2],
5       "hidden_dims": [
6           (128,64,32,16),(256,128,64,32),
7           (128,64,32),(256,128,64),
8           (64,32),(128,64),(256,128),
9           (128,),(64,)
10      ],
11      "weight_decay": [0.0, 0.00001, 0.0001, 0.001
           , 0.01],
12      }
```

## D.2 Details on Cross Validation

**L1: Random Splitting.** We first split $D$ into 10 disjoint subsets, and then rotate on which ones are $D_{dev}$ and $D_{test}$. To save computation budget, hyperparameter selection is done on the $D_{dev}$ of the first fold.

**L2.1/L2.2/L3: Random Splitting at Different Granularity.** To ensure that the results are comparable across settings as much as possible, we use

| Model | Runtime (second) |
|---|---|
| Matrix+Task Baseline | 4 |
| Adapted SVD | 2 |
| Model-model kNN | 2 |
| Task-task kNN | 5 |
| Random Forest | 98 |
| Gradient Boosted Trees | 242 |
| MLP | 86 |
| MLP (optimized for search) | 28 |

Table 5: Runtime (Training+Evaluation) of Performance Prediction Models.

10-fold cross validation for all these settings, similar to the practice in L1. This ensures for every fold, the sizes of $D_{train}/D_{dev}/D_{test}$ are consistent across settings, and each example in $D$ appears in $D_{test}$ exactly once. In this case, the only changing variable is the data splitting strategy.

**L3 Composition: Holding out $\mathcal{L}_{test} \times \mathcal{T}_{test}$.** To align with the 10-fold setting in L1-L3 as much as possible, we split $\mathcal{L}$ into 3 non-overlapping subsets, and split $\mathcal{T}$ into 3 non-overlapping subsets. This gives $3 \times 3 = 9$ different $D_{test}$ for cross-validation.

## D.3 Hardware and Runtime

Matrix completion experiments and tree-based experiments are done on a server with 56 Intel Xeon CPU E5-2690v4 (@ 2.6 GHz). MLP experiments are done on one NVIDIA GeForce RTX 2080 Ti. The runtime for training a model for performance prediction is listed in Table 5. For small-bench search, we optimized MLP training by not saving a copy of the best checkpoint in memory, and moving $D_{train}$ and $D_{dev}$ to GPU before training to reduce GPU I/O. For Greedy Search, the performance prediction algorithm is repeated for 12326 times, using approximately 5 days. For Best of 5000, the performance prediction algorithm is repeated for 5000 times, using approximately 2 days.

# E  Full Results of Performance Prediction

| | RMSE | $R^2$ Score | Task Avg. Pearson | Task Avg. Kendall |
|---|---|---|---|---|
| **L1 Setting: Random $(l, n_{param}, t, n_{shot})$, 10 folds** | | | | |
| Model+Task Baseline | 0.1220 ($\pm$ 0.0016) | 0.7068 ($\pm$ 0.0067) | 0.5207 ($\pm$ 0.0078) | 0.3617 ($\pm$ 0.0077) |
| Adapted SVD | 0.0986 ($\pm$ 0.0014) | 0.8082 ($\pm$ 0.0053) | 0.5149 ($\pm$ 0.0089) | 0.3443 ($\pm$ 0.0053) |
| Model-model kNN | 0.0712 ($\pm$ 0.0016) | 0.8999 ($\pm$ 0.0041) | 0.7046 ($\pm$ 0.0085) | 0.5133 ($\pm$ 0.0070) |
| Task-task kNN | 0.0695 ($\pm$ 0.0012) | 0.9048 ($\pm$ 0.0035) | 0.7006 ($\pm$ 0.0091) | 0.5109 ($\pm$ 0.0078) |
| Random Forest | 0.0553 ($\pm$ 0.0014) | 0.9397 ($\pm$ 0.0030) | 0.8378 ($\pm$ 0.0062) | 0.6590 ($\pm$ 0.0058) |
| Gradient Boosted Trees | 0.0499 ($\pm$ 0.0013) | 0.9510 ($\pm$ 0.0025) | 0.8566 ($\pm$ 0.0085) | 0.6690 ($\pm$ 0.0060) |
| MLP | 0.0500 ($\pm$ 0.0009) | 0.9508 ($\pm$ 0.0014) | 0.8203 ($\pm$ 0.0097) | 0.6128 ($\pm$ 0.0065) |
| **L2.1 Setting: Random $(l, n_{param}, t)$, 10 folds** | | | | |
| Model+Task Baseline | 0.1221 ($\pm$ 0.0042) | 0.7057 ($\pm$ 0.0158) | 0.4823 ($\pm$ 0.0172) | 0.3629 ($\pm$ 0.0152) |
| Adapted SVD | 0.1000 ($\pm$ 0.0039) | 0.8026 ($\pm$ 0.0135) | 0.4575 ($\pm$ 0.0115) | 0.3229 ($\pm$ 0.0121) |
| Model-model kNN | 0.0721 ($\pm$ 0.0034) | 0.8974 ($\pm$ 0.0085) | 0.6669 ($\pm$ 0.0118) | 0.5043 ($\pm$ 0.0110) |
| Task-task kNN | 0.0763 ($\pm$ 0.0031) | 0.8852 ($\pm$ 0.0078) | 0.6102 ($\pm$ 0.0190) | 0.4548 ($\pm$ 0.0136) |
| Random Forest | 0.0724 ($\pm$ 0.0034) | 0.8962 ($\pm$ 0.0109) | 0.7468 ($\pm$ 0.0152) | 0.5830 ($\pm$ 0.0101) |
| Gradient Boosted Trees | 0.0676 ($\pm$ 0.0029) | 0.9095 ($\pm$ 0.0084) | 0.7458 ($\pm$ 0.0155) | 0.5759 ($\pm$ 0.0094) |
| MLP | 0.0616 ($\pm$ 0.0022) | 0.9251 ($\pm$ 0.0040) | 0.7008 ($\pm$ 0.0163) | 0.5244 ($\pm$ 0.0126) |
| **L2.2 Setting: Random $(l, t, n_{shot})$, 10 folds** | | | | |
| Model+Task Baseline | 0.1315 ($\pm$ 0.0033) | 0.6577 ($\pm$ 0.0288) | 0.4556 ($\pm$ 0.0327) | 0.3373 ($\pm$ 0.0229) |
| Adapted SVD | 0.1208 ($\pm$ 0.0042) | 0.7107 ($\pm$ 0.0324) | 0.4338 ($\pm$ 0.0229) | 0.3095 ($\pm$ 0.0170) |
| Model-model kNN | 0.0983 ($\pm$ 0.0062) | 0.8072 ($\pm$ 0.0339) | 0.5885 ($\pm$ 0.0204) | 0.4287 ($\pm$ 0.0116) |
| Task-task kNN | 0.0985 ($\pm$ 0.0060) | 0.8079 ($\pm$ 0.0231) | 0.6059 ($\pm$ 0.0181) | 0.4449 ($\pm$ 0.0137) |
| Random Forest | 0.0675 ($\pm$ 0.0031) | 0.9098 ($\pm$ 0.0096) | 0.7571 ($\pm$ 0.0144) | 0.5862 ($\pm$ 0.0107) |
| Gradient Boosted Trees | 0.0624 ($\pm$ 0.0033) | 0.9229 ($\pm$ 0.0092) | 0.7819 ($\pm$ 0.0116) | 0.6057 ($\pm$ 0.0104) |
| MLP | 0.0555 ($\pm$ 0.0019) | 0.9391 ($\pm$ 0.0050) | 0.7424 ($\pm$ 0.0210) | 0.5520 ($\pm$ 0.0143) |
| **L3 Setting: Random $(l, t)$, 10 folds** | | | | |
| Model+Task Baseline | 0.1321 ($\pm$ 0.0075) | 0.6542 ($\pm$ 0.0302) | 0.4746 ($\pm$ 0.0288) | 0.3575 ($\pm$ 0.0229) |
| Adapted SVD | 0.1199 ($\pm$ 0.0088) | 0.7146 ($\pm$ 0.0353) | 0.4211 ($\pm$ 0.0291) | 0.2979 ($\pm$ 0.0211) |
| Model-model kNN | 0.1028 ($\pm$ 0.0130) | 0.7880 ($\pm$ 0.0531) | 0.5954 ($\pm$ 0.0214) | 0.4345 ($\pm$ 0.0106) |
| Task-task kNN | 0.1032 ($\pm$ 0.0076) | 0.7878 ($\pm$ 0.0308) | 0.5667 ($\pm$ 0.0242) | 0.4102 ($\pm$ 0.0156) |
| Random Forest | 0.1015 ($\pm$ 0.0103) | 0.7940 ($\pm$ 0.0414) | 0.6617 ($\pm$ 0.0234) | 0.4961 ($\pm$ 0.0174) |
| Gradient Boosted Trees | 0.0947 ($\pm$ 0.0084) | 0.8212 ($\pm$ 0.0315) | 0.6589 ($\pm$ 0.0302) | 0.4827 ($\pm$ 0.0246) |
| MLP | 0.0736 ($\pm$ 0.0064) | 0.8922 ($\pm$ 0.0164) | 0.6573 ($\pm$ 0.0156) | 0.4680 ($\pm$ 0.0148) |
| **L3 Composition Setting: Holding out $\mathcal{L}_{test} \times \mathcal{T}_{test}$, 9 folds** | | | | |
| Model+Task Baseline | 0.1383 ($\pm$ 0.0115) | 0.6231 ($\pm$ 0.0265) | 0.5099 ($\pm$ 0.0763) | 0.3709 ($\pm$ 0.0457) |
| Adapted SVD | 0.1318 ($\pm$ 0.0142) | 0.6575 ($\pm$ 0.0460) | 0.4720 ($\pm$ 0.0767) | 0.3365 ($\pm$ 0.0541) |
| Model-model kNN | 0.1138 ($\pm$ 0.0173) | 0.7398 ($\pm$ 0.0731) | 0.5342 ($\pm$ 0.1198) | 0.3918 ($\pm$ 0.0956) |
| Task-task kNN | 0.1152 ($\pm$ 0.0139) | 0.7362 ($\pm$ 0.0530) | 0.5692 ($\pm$ 0.0773) | 0.4086 ($\pm$ 0.0657) |
| Random Forest | 0.1118 ($\pm$ 0.0154) | 0.7475 ($\pm$ 0.0782) | 0.6136 ($\pm$ 0.0781) | 0.4347 ($\pm$ 0.0494) |
| Gradient Boosted Trees | 0.1072 ($\pm$ 0.0104) | 0.7706 ($\pm$ 0.0451) | 0.5970 ($\pm$ 0.0941) | 0.4034 ($\pm$ 0.0588) |
| MLP | 0.0843 ($\pm$ 0.0072) | 0.8597 ($\pm$ 0.0129) | 0.6228 ($\pm$ 0.0991) | 0.4236 ($\pm$ 0.0675) |

Table 6: **Full Results of Performance Prediction.** ■ Matrix Completion ■ Trees ■ Neural Network

## F    Scaling Behavior of Most/Least "Predictable" Tasks (§3.4)

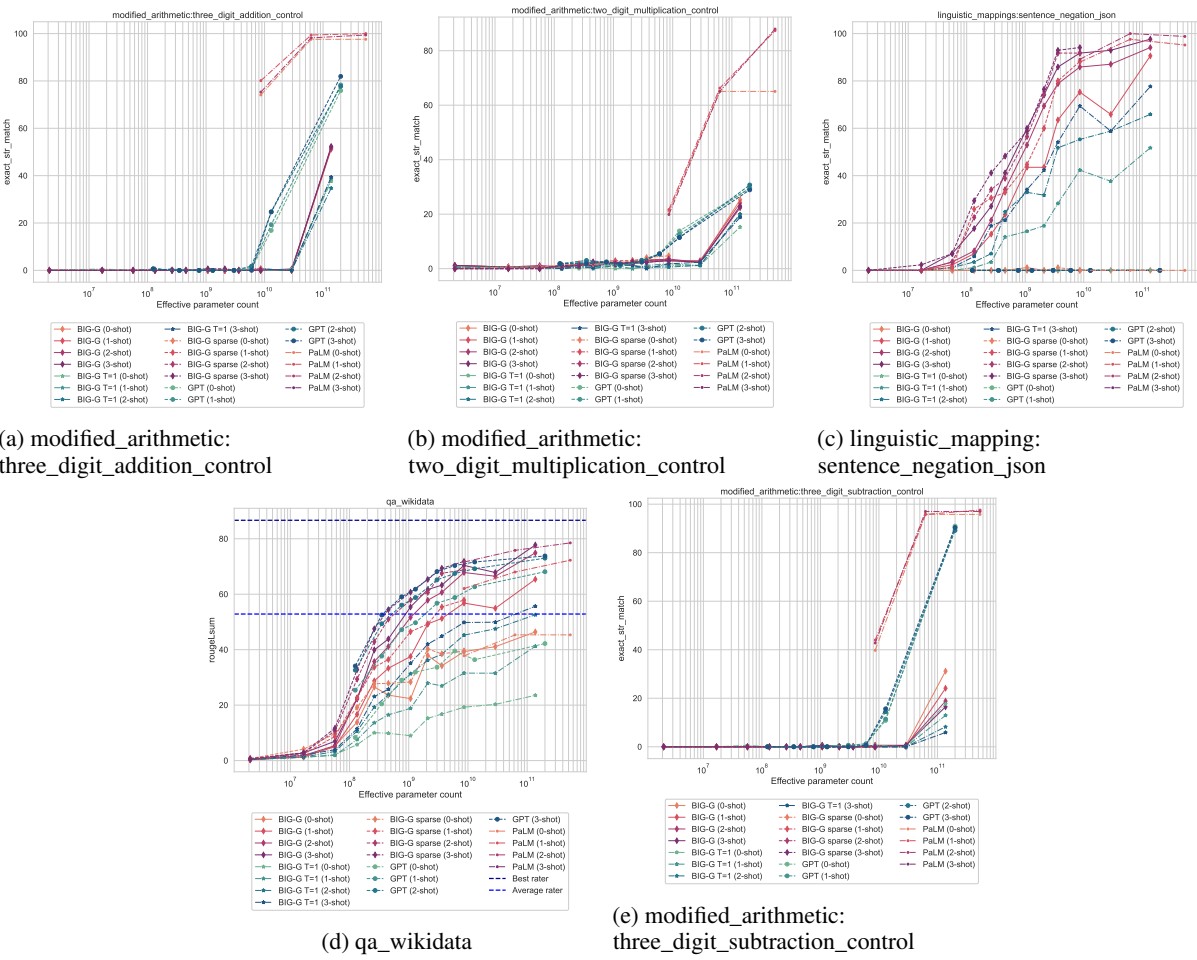

(a) modified_arithmetic:
three_digit_addition_control

(b) modified_arithmetic:
two_digit_multiplication_control

(c) linguistic_mapping:
sentence_negation_json

(d) qa_wikidata

(e) modified_arithmetic:
three_digit_subtraction_control

Figure 9: **Scaling Behavior of Tasks with Highest** $R^2$ **Score.** Figures are obtained from https://github.com/google/BIG-bench

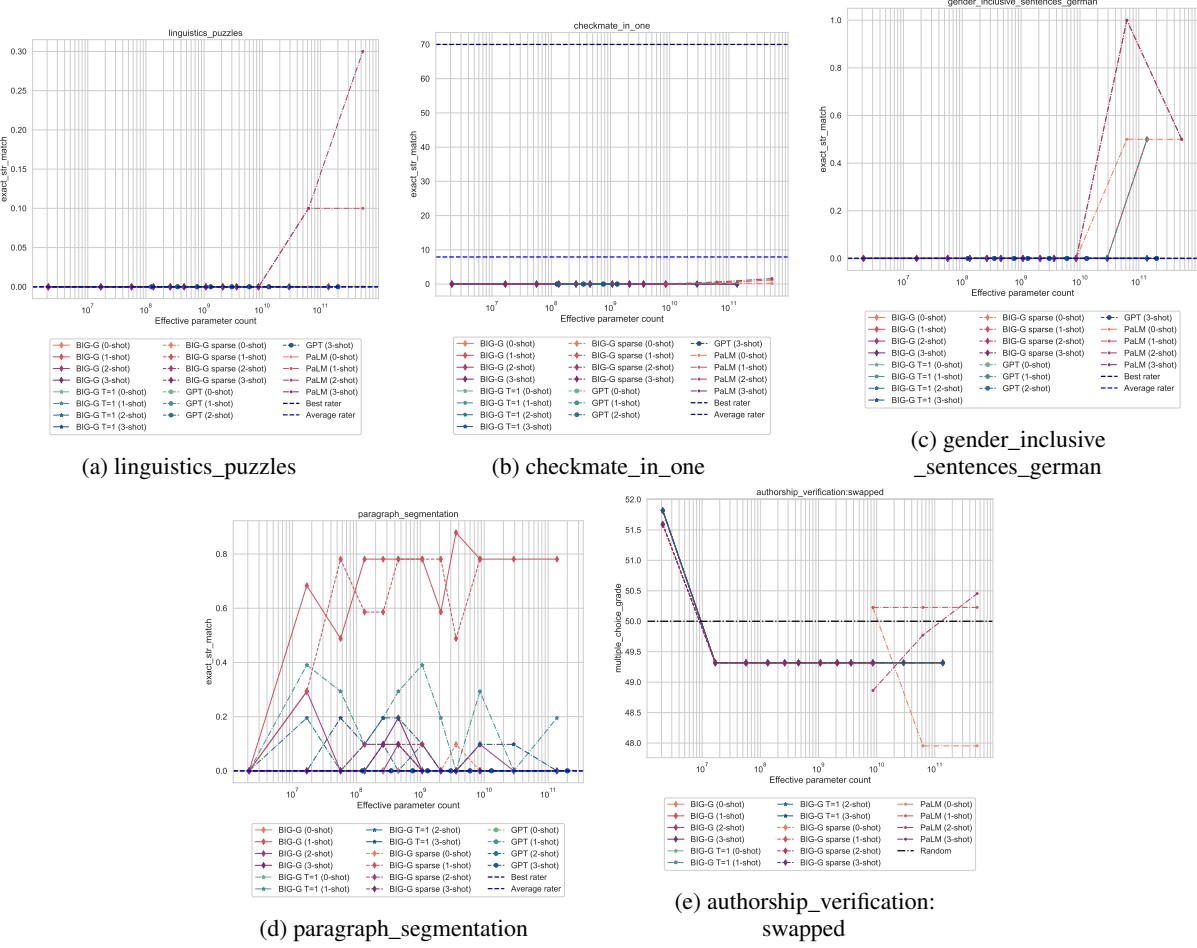

(a) linguistics_puzzles

(b) checkmate_in_one

(c) gender_inclusive
_sentences_german

(d) paragraph_segmentation

(e) authorship_verification:
swapped

Figure 10: **Scaling Behavior of Tasks with Lowest $R^2$ Score.** Figures are obtained from `https://github.com/google/BIG-bench`

# G "Small-bench" Search Results

## G.1 BIG-bench Hard and BIG-bench Lite

The two subsets listed below slightly differ from the original subsets, because we filtered out several subtasks as described in §A.

```
1  {
2      "bblite": ['bbq_lite_json:bbq_lite_json_age_ambig', 'bbq_lite_json:bbq_lite_json_age_disambig', '
           bbq_lite_json:bbq_lite_json_disability_status_ambig', 'bbq_lite_json:
           bbq_lite_json_disability_status_disambig', 'bbq_lite_json:bbq_lite_json_gender_identity_ambig',
            'bbq_lite_json:bbq_lite_json_gender_identity_disambig', 'bbq_lite_json:
           bbq_lite_json_nationality_ambig', 'bbq_lite_json:bbq_lite_json_nationality_disambig', '
           bbq_lite_json:bbq_lite_json_physical_appearance_ambig', 'bbq_lite_json:
           bbq_lite_json_physical_appearance_disambig', 'bbq_lite_json:bbq_lite_json_race_ethnicity_ambig'
           , 'bbq_lite_json:bbq_lite_json_race_ethnicity_disambig', 'bbq_lite_json:
           bbq_lite_json_religion_ambig', 'bbq_lite_json:bbq_lite_json_religion_disambig', 'bbq_lite_json:
           bbq_lite_json_ses_ambig', 'bbq_lite_json:bbq_lite_json_ses_disambig', 'bbq_lite_json:
           bbq_lite_json_sexual_orientation_ambig', 'bbq_lite_json:
           bbq_lite_json_sexual_orientation_disambig', 'code_line_description', 'conceptual_combinations:
           emergent_properties', 'formal_fallacies_syllogisms_negation', 'hindu_knowledge', '
           language_identification', 'linguistics_puzzles', 'logic_grid_puzzle', 'logical_deduction:
           five_objects', 'logical_deduction:seven_objects', 'logical_deduction:three_objects', '
           novel_concepts', 'operators', 'parsinlu_reading_comprehension', 'play_dialog_same_or_different'
           , 'strange_stories:boolean', 'strange_stories:multiple_choice', 'strategyqa', '
           symbol_interpretation:adversarial', 'symbol_interpretation:emoji_agnostic', '
           symbol_interpretation:name_agnostic', 'symbol_interpretation:plain', 'symbol_interpretation:
           tricky', 'vitaminc_fact_verification', 'winowhy'],
3      "bbhard": ['causal_judgment', 'date_understanding', 'disambiguation_qa', 'dyck_languages', '
           formal_fallacies_syllogisms_negation', 'geometric_shapes', 'hyperbaton', 'logical_deduction:
           five_objects', 'logical_deduction:seven_objects', 'logical_deduction:three_objects', '
           movie_recommendation', 'navigate', 'object_counting', 'penguins_in_a_table', '
           reasoning_about_colored_objects', 'ruin_names', 'salient_translation_error_detection', 'snarks'
           , 'sports_understanding', 'temporal_sequences', 'tracking_shuffled_objects:five_objects', '
           tracking_shuffled_objects:seven_objects', 'tracking_shuffled_objects:three_objects', '
           word_sorting']
4  }
```

## G.2 Best of 5000

```
1  {
2      4: ['arithmetic:5_digit_subtraction', 'goal_step_wikihow:goal_inference', 'linguistic_mappings:
           past_tense_irregular_json', 'natural_instructions:subtask027_drop_answer_type_generation'],
3      8: ['arithmetic:4_digit_multiplication', 'chess_state_tracking:real_medium', 'elementary_math_qa:
           question_only', 'linguistic_mappings:past_tense_regular_json', 'natural_instructions:subtask015
           _mctaco_question_generation_frequency', 'natural_instructions:subtask034
           _winogrande_question_modification_object', 'natural_instructions:subtask038_qasc_combined_fact'
           , 'physics'],
4      16: ['arithmetic:2_digit_addition', 'arithmetic:3_digit_multiplication', 'arithmetic:4
           _digit_subtraction', 'bbq_lite_json:bbq_lite_json_gender_identity_ambig', 'bbq_lite_json:
           bbq_lite_json_physical_appearance_disambig', 'cause_and_effect:two_sentences', '
           chess_state_tracking:real_medium', 'chess_state_tracking:synthetic_medium', 'gem:cs_restaurants
           ', 'goal_step_wikihow:goal_inference', 'international_phonetic_alphabet_nli', '
           linguistic_mappings:plural_regular_json', 'natural_instructions:subtask005
           _mctaco_wrong_answer_generation_event_duration', 'natural_instructions:subtask036
           _qasc_topic_word_to_generate_related_fact', 'natural_instructions:subtask055
           _multirc_write_incorrect_answer', 'social_iqa'],
5      24: ['anachronisms', 'arithmetic:2_digit_division', 'arithmetic:3_digit_division', 'arithmetic:4
           _digit_subtraction', 'bbq_lite_json:bbq_lite_json_physical_appearance_ambig', 'dyck_languages',
            'elementary_math_qa:question_with_language_hint', 'gem:asset', 'implicatures', '
           linguistic_mappings:plural_json', 'mathematical_induction', 'mnist_ascii', '
           natural_instructions:subtask014_mctaco_wrong_answer_generation_absolute_timepoint', '
           natural_instructions:subtask017_mctaco_wrong_answer_generation_frequency', '
           natural_instructions:subtask044_essential_terms_identifying_essential_words', '
           natural_instructions:subtask060_ropes_question_generation', 'natural_instructions:subtask061
           _ropes_answer_generation', 'navigate', 'nonsense_words_grammar', 'persian_idioms', 'social_iqa'
           , 'strange_stories:multiple_choice', 'unnatural_in_context_learning:dates', 'winowhy'],
6      32: ['abstract_narrative_understanding:4_distractors', 'arithmetic:2_digit_division', 'arithmetic:2
           _digit_subtraction', 'arithmetic:3_digit_addition', 'arithmetic:3_digit_division', 'arithmetic:
           4_digit_division', 'arithmetic:5_digit_addition', 'bbq_lite_json:
           bbq_lite_json_gender_identity_disambig', 'bridging_anaphora_resolution_barqa', '
           chess_state_tracking:synthetic_long', 'common_morpheme', 'elementary_math_qa:
           question_with_mathematical_hint', 'epistemic_reasoning', 'gem:cs_restaurants', '
           identify_math_theorems', 'implicatures', 'intersect_geometry:shapes_4', 'kannada', '
           linguistic_mappings:past_tense_json', 'linguistic_mappings:plural_json', 'logical_deduction:
           three_objects', 'natural_instructions:subtask005_mctaco_wrong_answer_generation_event_duration'
           , 'natural_instructions:subtask013_mctaco_answer_generation_absolute_timepoint', '
           natural_instructions:subtask020_mctaco_span_based_question', 'natural_instructions:subtask023
           _cosmosqa_question_generation', 'natural_instructions:subtask035
           _winogrande_question_modification_person', 'natural_instructions:subtask045
           _miscellaneous_sentence_paraphrasing', 'natural_instructions:subtask054
           _multirc_write_correct_answer', 'natural_instructions:subtask056
           _multirc_classify_correct_answer', 'unit_interpretation:lv0', 'unnatural_in_context_learning:
           dates_unnatural_content', 'unnatural_in_context_learning:reverse_to_natural_content'],
```

```
7        42: ['abstract_narrative_understanding:9_distractors', 'analogical_similarity', 'arithmetic:2
            _digit_multiplication', 'arithmetic:2_digit_subtraction', 'arithmetic:3_digit_division', '
            arithmetic:5_digit_addition', 'bbq_lite_json:bbq_lite_json_age_disambig', 'cause_and_effect:
            two_sentences', 'color:hsl', 'cs_algorithms:lcs', 'cs_algorithms:valid_parentheses', '
            elementary_math_qa:language_hint_only', 'english_russian_proverbs', 'fantasy_reasoning', 'gem:
            webnlg_en', 'gem:webnlg_ru', 'human_organs_senses', 'intent_recognition', 'intersect_geometry:
            shapes_3', 'linguistic_mappings:plural_json', 'metaphor_understanding:met2lit', 'mnist_ascii',
            'modified_arithmetic:three_digit_addition_control', 'movie_recommendation', '
            natural_instructions:subtask007_mctaco_answer_generation_transient_stationary', '
            natural_instructions:subtask016_mctaco_answer_generation_frequency', 'natural_instructions:
            subtask024_cosmosqa_answer_generation', 'natural_instructions:subtask026
            _drop_question_generation', 'natural_instructions:subtask034
            _winogrande_question_modification_object', 'natural_instructions:subtask037
            _qasc_generate_related_fact', 'natural_instructions:subtask047_misc_answering_science_questions
            ', 'natural_instructions:subtask049_multirc_questions_needed_to_answer', '
            paragraph_segmentation', 'question_selection', 'simple_ethical_questions', 'social_iqa', '
            strange_stories:boolean', 'swedish_to_german_proverbs', 'symbol_interpretation:tricky', '
            tracking_shuffled_objects:five_objects', 'unit_interpretation:lv3', '
            unnatural_in_context_learning:unnatural_addition_2_digit'],
8    }
```

## G.3 Greedy Search

```
1    {
2        4: ['gem:cs_restaurants', 'unnatural_in_context_learning:dates_unnatural_content', '
            natural_instructions:subtask058_multirc_question_answering', 'presuppositions_as_nli'],
3        8: ['gem:cs_restaurants', 'unnatural_in_context_learning:dates_unnatural_content', '
            natural_instructions:subtask058_multirc_question_answering', 'presuppositions_as_nli', '
            arithmetic:5_digit_division', 'evaluating_information_essentiality', 'simp_turing_concept:
            additional_set_2', 'linguistic_mappings:plural_regular_json'],
4        16: ['gem:cs_restaurants', 'unnatural_in_context_learning:dates_unnatural_content', '
            natural_instructions:subtask058_multirc_question_answering', 'presuppositions_as_nli', '
            arithmetic:5_digit_division', 'evaluating_information_essentiality', 'simp_turing_concept:
            additional_set_2', 'linguistic_mappings:plural_regular_json', 'gem:wikilingua_en', 'gem:
            common_gen', 'arithmetic:5_digit_subtraction', 'ruin_names', 'gem:asset', 'unit_interpretation:
            lv2', 'bbq_lite_json:bbq_lite_json_disability_status_ambig', 'physics'],
5        24: ['gem:cs_restaurants', 'unnatural_in_context_learning:dates_unnatural_content', '
            natural_instructions:subtask058_multirc_question_answering', 'presuppositions_as_nli', '
            arithmetic:5_digit_division', 'evaluating_information_essentiality', 'simp_turing_concept:
            additional_set_2', 'linguistic_mappings:plural_regular_json', 'gem:wikilingua_en', 'gem:
            common_gen', 'arithmetic:5_digit_subtraction', 'ruin_names', 'gem:asset', 'unit_interpretation:
            lv2', 'bbq_lite_json:bbq_lite_json_disability_status_ambig', 'physics', 'real_or_fake_text:easy
            ', 'strange_stories:multiple_choice', 'penguins_in_a_table', 'nonsense_words_grammar', '
            conceptual_combinations:emergent_properties', 'code_line_description', 'logical_sequence', '
            hhh_alignment:Helpfulness']
6        32: ['gem:cs_restaurants', 'unnatural_in_context_learning:dates_unnatural_content', '
            natural_instructions:subtask058_multirc_question_answering', 'presuppositions_as_nli', '
            arithmetic:5_digit_division', 'evaluating_information_essentiality', 'simp_turing_concept:
            additional_set_2', 'linguistic_mappings:plural_regular_json', 'gem:wikilingua_en', 'gem:
            common_gen', 'arithmetic:5_digit_subtraction', 'ruin_names', 'gem:asset', 'unit_interpretation:
            lv2', 'bbq_lite_json:bbq_lite_json_disability_status_ambig', 'physics', 'real_or_fake_text:easy
            ', 'strange_stories:multiple_choice', 'penguins_in_a_table', 'nonsense_words_grammar', '
            conceptual_combinations:emergent_properties', 'code_line_description', 'logical_sequence', '
            hhh_alignment:Helpfulness', 'linguistic_mappings:pronoun_replacement_json', 'gem:xsum', '
            cs_algorithms:lcs', 'chess_state_tracking:real_long', 'unnatural_in_context_learning:dates', '
            entailed_polarity_hindi', 'bbq_lite_json:bbq_lite_json_nationality_ambig', '
            natural_instructions:subtask023_cosmosqa_question_generation'],
7        42: ['gem:cs_restaurants', 'unnatural_in_context_learning:dates_unnatural_content', '
            natural_instructions:subtask058_multirc_question_answering', 'presuppositions_as_nli', '
            arithmetic:5_digit_division', 'evaluating_information_essentiality', 'simp_turing_concept:
            additional_set_2', 'linguistic_mappings:plural_regular_json', 'gem:wikilingua_en', 'gem:
            common_gen', 'arithmetic:5_digit_subtraction', 'ruin_names', 'gem:asset', 'unit_interpretation:
            lv2', 'bbq_lite_json:bbq_lite_json_disability_status_ambig', 'physics', 'real_or_fake_text:easy
            ', 'strange_stories:multiple_choice', 'penguins_in_a_table', 'nonsense_words_grammar', '
            conceptual_combinations:emergent_properties', 'code_line_description', 'logical_sequence', '
            hhh_alignment:Helpfulness', 'linguistic_mappings:pronoun_replacement_json', 'gem:xsum', '
            cs_algorithms:lcs', 'chess_state_tracking:real_long', 'unnatural_in_context_learning:dates', '
            entailed_polarity_hindi', 'bbq_lite_json:bbq_lite_json_nationality_ambig', '
            natural_instructions:subtask023_cosmosqa_question_generation', 'natural_instructions:subtask004
            _mctaco_answer_generation_event_duration', 'question_selection', 'simp_turing_concept:
            additional_set_1', 'matrixshapes', 'movie_dialog_same_or_different', 'natural_instructions:
            subtask021_mctaco_grammatical_logical', 'arithmetic:4_digit_division', 'natural_instructions:
            subtask061_ropes_answer_generation', 'modified_arithmetic:three_digit_subtraction_control', '
            date_understanding']
8    }
```

## G.4 Beam Search (q=4, p=1/4)

```
1    {
2        4: ['implicatures', 'natural_instructions:subtask049_multirc_questions_needed_to_answer', '
            linguistic_mappings:past_tense_json', 'natural_instructions:subtask035
            _winogrande_question_modification_person'],
```

```
3      8: ['implicatures', 'natural_instructions:subtask049_multirc_questions_needed_to_answer', '
          linguistic_mappings:past_tense_json', 'natural_instructions:subtask057
          _multirc_classify_incorrect_answer', 'gem:e2e_nlg', 'arithmetic:5_digit_subtraction', '
          logical_deduction:seven_objects', 'bbq_lite_json:bbq_lite_json_nationality_ambig'],
4      16: ['implicatures', 'natural_instructions:subtask049_multirc_questions_needed_to_answer', '
          linguistic_mappings:past_tense_json', 'natural_instructions:subtask057
          _multirc_classify_incorrect_answer', 'gem:e2e_nlg', 'arithmetic:5_digit_subtraction', '
          logical_deduction:seven_objects', 'cryobiology_spanish', 'vitaminc_fact_verification', '
          qa_wikidata', 'abstract_narrative_understanding:4_distractors', 'linguistic_mappings:
          plural_json', 'symbol_interpretation:emoji_agnostic', 'elementary_math_qa:
          question_with_mathematical_hint', 'simp_turing_concept:additional_set_2', 'elementary_math_qa:
          question_only'],
5      24: ['implicatures', 'natural_instructions:subtask049_multirc_questions_needed_to_answer', '
          linguistic_mappings:past_tense_json', 'natural_instructions:subtask057
          _multirc_classify_incorrect_answer', 'gem:e2e_nlg', 'arithmetic:5_digit_subtraction', '
          logical_deduction:seven_objects', 'cryobiology_spanish', 'vitaminc_fact_verification', '
          qa_wikidata', 'abstract_narrative_understanding:4_distractors', 'linguistic_mappings:
          plural_json', 'symbol_interpretation:emoji_agnostic', 'elementary_math_qa:
          question_with_mathematical_hint', 'simp_turing_concept:additional_set_2', 'elementary_math_qa:
          question_only', 'natural_instructions:subtask004_mctaco_answer_generation_event_duration', '
          empirical_judgments', 'gre_reading_comprehension', 'logical_deduction:three_objects', '
          arithmetic:4_digit_subtraction', 'hhh_alignment:Harms', 'bbq_lite_json:
          bbq_lite_json_nationality_ambig', 'natural_instructions:subtask025
          _cosmosqa_incorrect_answer_generation'],
6      32: ['implicatures', 'natural_instructions:subtask049_multirc_questions_needed_to_answer', '
          linguistic_mappings:past_tense_json', 'natural_instructions:subtask057
          _multirc_classify_incorrect_answer', 'gem:e2e_nlg', 'arithmetic:5_digit_subtraction', '
          logical_deduction:seven_objects', 'cryobiology_spanish', 'vitaminc_fact_verification', '
          qa_wikidata', 'abstract_narrative_understanding:4_distractors', 'linguistic_mappings:
          plural_json', 'symbol_interpretation:emoji_agnostic', 'elementary_math_qa:
          question_with_mathematical_hint', 'simp_turing_concept:additional_set_2', 'elementary_math_qa:
          question_only', 'natural_instructions:subtask004_mctaco_answer_generation_event_duration', '
          empirical_judgments', 'gre_reading_comprehension', 'logical_deduction:three_objects', '
          elementary_math_qa:language_hint_only', 'cifar10_classification:base64', '
          mathematical_induction', 'arithmetic:2_digit_subtraction', 'unnatural_in_context_learning:
          identity', 'natural_instructions:subtask054_multirc_write_correct_answer', 'logical_sequence', '
          'arithmetic:5_digit_multiplication', 'persian_idioms', 'linguistic_mappings:
          past_tense_regular_json', 'periodic_elements:subtask_1', 'parsinlu_reading_comprehension'],
7      42: ['implicatures', 'natural_instructions:subtask049_multirc_questions_needed_to_answer', '
          linguistic_mappings:past_tense_json', 'natural_instructions:subtask057
          _multirc_classify_incorrect_answer', 'gem:e2e_nlg', 'arithmetic:5_digit_subtraction', '
          logical_deduction:seven_objects', 'cryobiology_spanish', 'vitaminc_fact_verification', '
          qa_wikidata', 'abstract_narrative_understanding:4_distractors', 'linguistic_mappings:
          plural_json', 'symbol_interpretation:emoji_agnostic', 'elementary_math_qa:
          question_with_mathematical_hint', 'simp_turing_concept:additional_set_2', 'elementary_math_qa:
          question_only', 'natural_instructions:subtask004_mctaco_answer_generation_event_duration', '
          empirical_judgments', 'gre_reading_comprehension', 'logical_deduction:three_objects', '
          elementary_math_qa:language_hint_only', 'cifar10_classification:base64', '
          mathematical_induction', 'arithmetic:2_digit_subtraction', 'unnatural_in_context_learning:
          identity', 'natural_instructions:subtask054_multirc_write_correct_answer', 'logical_sequence', '
          'arithmetic:5_digit_multiplication', 'natural_instructions:subtask055
          _multirc_write_incorrect_answer', 'natural_instructions:subtask034
          _winogrande_question_modification_object', 'arithmetic:3_digit_addition', 'natural_instructions
          :subtask016_mctaco_answer_generation_frequency', 'language_identification', 'crass_ai', '
          natural_instructions:subtask050_multirc_answerability', 'persian_idioms', 'disambiguation_qa', '
          'gem:asset', 'dark_humor_detection', 'arithmetic:4_digit_addition', '
          international_phonetic_alphabet_nli', 'natural_instructions:subtask008
          _mctaco_wrong_answer_generation_transient_stationary']
8  }
```

## G.5  $k$-means

The numbers reported in Fig. 6 is the average of 5 runs. We ran the $k$-means algorithms with 5 different random initialization. In the following we list the "small-bench" candidates from 1 run. The same applies to "$k$-means + Task Value" results.

```
1  {
2      4: ['anachronisms', 'natural_instructions:subtask032_winogrande_question_generation_person', '
          linguistic_mappings:de_past_tense_regular_json','natural_instructions:subtask033
          _winogrande_answer_generation']",
3      8: ['anachronisms', 'natural_instructions:subtask026_drop_question_generation', 'arithmetic:5
          _digit_addition', 'linguistic_mappings:question_formation_json', 'natural_instructions:subtask0
          30_winogrande_full_person', 'bbq_lite_json:bbq_lite_json_ses_disambig', 'natural_instructions:
          subtask039_qasc_find_overlapping_words', 'linguistic_mappings:de_past_tense_regular_json']",
4      16: ['anachronisms', 'symbol_interpretation:plain', 'arithmetic:5_digit_addition', '
          linguistic_mappings:past_tense_regular_json', 'natural_instructions:subtask031
          _winogrande_question_generation_object', 'bbq_lite_json:bbq_lite_json_ses_disambig', '
          natural_instructions:subtask039_qasc_find_overlapping_words', 'modified_arithmetic:
          two_digit_multiplication_plus_one', 'bbq_lite_json:bbq_lite_json_nationality_ambig', '
          chess_state_tracking:real_medium', 'linguistic_mappings:de_past_tense_regular_json', '
          natural_instructions:subtask032_winogrande_question_generation_person', 'natural_instructions:
          subtask041_qasc_answer_generation', 'gem:turk', 'unnatural_in_context_learning:
          dates_unnatural_content', 'natural_instructions:subtask048_multirc_question_generation']",
```

```
5       24: ['anachronisms', 'real_or_fake_text:easy', 'arithmetic:5_digit_addition', 'gem:webnlg_en', '
            natural_instructions:subtask049_multirc_questions_needed_to_answer', 'linguistic_mappings:
            past_tense_regular_json', 'bbq_lite_json:bbq_lite_json_nationality_ambig', 'gem:xsum', '
            tracking_shuffled_objects:seven_objects', 'natural_instructions:subtask019
            _mctaco_temporal_reasoning_category', 'natural_instructions:subtask024
            _cosmosqa_answer_generation', 'natural_instructions:subtask030_winogrande_full_person', '
            chess_state_tracking:synthetic_medium', 'natural_instructions:subtask045
            _miscellaneous_sentence_paraphrasing', 'unnatural_in_context_learning:
            reverse_to_natural_content', 'authorship_verification:swapped', 'natural_instructions:subtask03
            9_qasc_find_overlapping_words', 'modified_arithmetic:two_digit_multiplication_control', '
            bbq_lite_json:bbq_lite_json_sexual_orientation_disambig', 'abstract_narrative_understanding:9
            _distractors', 'color:hsl', 'fact_checker:fever', 'tense', 'intersect_geometry:shapes_4']",
6       32: ['anachronisms', 'real_or_fake_text:easy', 'arithmetic:5_digit_division', 'gem:webnlg_en', '
            natural_instructions:subtask049_multirc_questions_needed_to_answer', 'linguistic_mappings:
            past_tense_json', 'bbq_lite_json:bbq_lite_json_nationality_ambig', 'gem:xsum', '
            tracking_shuffled_objects:seven_objects', 'natural_instructions:subtask020
            _mctaco_span_based_question', 'natural_instructions:subtask032
            _winogrande_question_generation_person', 'natural_instructions:subtask032
            _winogrande_question_generation_person', 'natural_instructions:subtask003
            _mctaco_question_generation_event_duration', 'natural_instructions:subtask045
            _miscellaneous_sentence_paraphrasing', 'unnatural_in_context_learning:reverse_natural_content',
            'authorship_verification:swapped', 'goal_step_wikihow:goal_inference', 'modified_arithmetic:
            two_digit_multiplication_control', 'bbq_lite_json:bbq_lite_json_sexual_orientation_disambig', '
            abstract_narrative_understanding:9_distractors', 'color:hsl', 'fact_checker:fever', 'tense', '
            intersect_geometry:shapes_4', 'unnatural_in_context_learning:dates_unnatural_content', '
            natural_instructions:subtask048_multirc_question_generation', 'hyperbaton', '
            symbol_interpretation:plain', 'arithmetic:5_digit_addition', 'chess_state_tracking:real_medium'
            , 'simp_turing_concept:additional_set_1', 'physics']",
7       42: ['anachronisms', 'real_or_fake_text:easy', 'arithmetic:5_digit_division', 'gem:webnlg_en', '
            natural_instructions:subtask039_qasc_find_overlapping_words', 'linguistic_mappings:
            past_tense_json', 'bbq_lite_json:bbq_lite_json_nationality_ambig', 'gem:xsum', '
            tracking_shuffled_objects:seven_objects', 'natural_instructions:subtask019
            _mctaco_temporal_reasoning_category', 'natural_instructions:subtask017
            _mctaco_wrong_answer_generation_frequency', 'natural_instructions:subtask030
            _winogrande_full_person', 'natural_instructions:subtask003
            _mctaco_question_generation_event_duration', 'natural_instructions:subtask045
            _miscellaneous_sentence_paraphrasing', 'unnatural_in_context_learning:reverse_natural_content',
            'authorship_verification:swapped', 'goal_step_wikihow:goal_inference', 'modified_arithmetic:
            two_digit_multiplication_control', 'bbq_lite_json:bbq_lite_json_physical_appearance_disambig',
            'abstract_narrative_understanding:9_distractors', 'color:hex', 'fact_checker:fever', 'tense', '
            linguistic_mappings:de_past_tense_regular_json', 'unnatural_in_context_learning:
            dates_unnatural_content', 'natural_instructions:subtask026_drop_question_generation', '
            hyperbaton', 'symbol_interpretation:plain', 'arithmetic:5_digit_addition', '
            chess_state_tracking:real_medium', 'simp_turing_concept:additional_set_1', 'physics', 'cifar10
            _classification:base64', 'intersect_geometry:shapes_4', 'elementary_math_qa:
            question_with_language_hint', 'goal_step_wikihow:step_inference', 'unit_conversion:
            unit_identification', 'natural_instructions:subtask056_multirc_classify_correct_answer', '
            strange_stories:multiple_choice', 'discourse_marker_prediction', 'bbq_lite_json:
            bbq_lite_json_religion_disambig', 'natural_instructions:subtask024_cosmosqa_answer_generation']
            "
8    }
```

## G.6  *k*-means + Task Value

```
1   {
2       4: ['bbq_lite_json:bbq_lite_json_sexual_orientation_disambig', 'natural_instructions:subtask029
            _winogrande_full_object', 'gem:xsum', 'linguistic_mappings:question_formation_json']",
3       8: ['hyperbaton', 'natural_instructions:subtask026_drop_question_generation', 'arithmetic:5
            _digit_addition', 'linguistic_mappings:question_formation_json', 'natural_instructions:subtask0
            29_winogrande_full_object', 'bbq_lite_json:bbq_lite_json_sexual_orientation_disambig', '
            natural_instructions:subtask003_mctaco_question_generation_event_duration', 'gem:xsum']",
4       16: ['hyperbaton', 'real_or_fake_text:gpt2_xl', 'arithmetic:5_digit_addition', 'linguistic_mappings:
            past_tense_regular_json', 'natural_instructions:subtask029_winogrande_full_object', '
            bbq_lite_json:bbq_lite_json_sexual_orientation_disambig', 'goal_step_wikihow:goal_inference', '
            periodic_elements:subtask_1', 'bbq_lite_json:bbq_lite_json_physical_appearance_ambig', '
            chess_state_tracking:synthetic_long', 'intersect_geometry:shapes_4', 'natural_instructions:
            subtask025_cosmosqa_incorrect_answer_generation', 'natural_instructions:subtask002
            _quoref_answer_generation', 'gem:turk', 'unnatural_in_context_learning:dates_unnatural_content'
            , 'natural_instructions:subtask048_multirc_question_generation']",
5       24: ['snarks', 'real_or_fake_text:gpt2_xl', 'arithmetic:5_digit_addition', 'simp_turing_concept:
            additional_set_2', 'natural_instructions:subtask048_multirc_question_generation', '
            linguistic_mappings:past_tense_regular_json', 'bbq_lite_json:
            bbq_lite_json_physical_appearance_ambig', 'gem:xsum', 'analogical_similarity', '
            natural_instructions:subtask019_mctaco_temporal_reasoning_category', 'natural_instructions:
            subtask025_cosmosqa_incorrect_answer_generation', 'natural_instructions:subtask029
            _winogrande_full_object', 'chess_state_tracking:real_short', 'natural_instructions:subtask045
            _miscellaneous_sentence_paraphrasing', 'unnatural_in_context_learning:
            reverse_to_natural_content', 'hyperbaton', 'goal_step_wikihow:goal_inference', '
            modified_arithmetic:two_digit_multiplication_control', 'bbq_lite_json:
            bbq_lite_json_sexual_orientation_disambig', 'nonsense_words_grammar', '
            discourse_marker_prediction', 'implicatures', 'ascii_word_recognition', 'intersect_geometry:
            shapes_4']",
6       32: ['anachronisms', 'real_or_fake_text:gpt2_xl', 'arithmetic:3_digit_division', '
            play_dialog_same_or_different', 'natural_instructions:subtask058_multirc_question_answering', '
            linguistic_mappings:past_tense_json', 'bbq_lite_json:bbq_lite_json_physical_appearance_ambig',
```

```
            'gem:xsum', 'analogical_similarity', 'natural_instructions:subtask020
            _mctaco_span_based_question', 'natural_instructions:subtask025
            _cosmosqa_incorrect_answer_generation', 'minute_mysteries_qa:multiplechoice', '
            natural_instructions:subtask003_mctaco_question_generation_event_duration', '
            natural_instructions:subtask045_miscellaneous_sentence_paraphrasing', '
            unnatural_in_context_learning:reverse_to_natural_content', 'authorship_verification:swapped', '
            goal_step_wikihow:goal_inference', 'modified_arithmetic:two_digit_multiplication_control', '
            bbq_lite_json:bbq_lite_json_sexual_orientation_disambig', 'figure_of_speech_detection', '
            discourse_marker_prediction', 'implicatures', 'ascii_word_recognition', 'intersect_geometry:
            shapes_4', 'unnatural_in_context_learning:dates_unnatural_content', 'natural_instructions:
            subtask048_multirc_question_generation', 'hyperbaton', 'symbol_interpretation:plain', '
            arithmetic:5_digit_addition', 'chess_state_tracking:synthetic_long', 'simp_turing_concept:
            additional_set_1', 'human_organs_senses']",
7       42: ['anachronisms', 'real_or_fake_text:gpt2_xl', 'arithmetic:3_digit_division', '
            play_dialog_same_or_different', 'natural_instructions:subtask047
            _misc_answering_science_questions', 'linguistic_mappings:past_tense_json', 'bbq_lite_json:
            bbq_lite_json_physical_appearance_ambig', 'gem:xsum', 'tracking_shuffled_objects:seven_objects'
            , 'natural_instructions:subtask019_mctaco_temporal_reasoning_category', 'natural_instructions:
            subtask017_mctaco_wrong_answer_generation_frequency', 'natural_instructions:subtask029
            _winogrande_full_object', 'natural_instructions:subtask003
            _mctaco_question_generation_event_duration', 'natural_instructions:subtask045
            _miscellaneous_sentence_paraphrasing', 'unnatural_in_context_learning:
            reverse_to_natural_content', 'authorship_verification:swapped', 'goal_step_wikihow:
            goal_inference', 'modified_arithmetic:two_digit_multiplication_control', 'bbq_lite_json:
            bbq_lite_json_physical_appearance_disambig', 'figure_of_speech_detection', 'color:hex', '
            implicatures', 'ascii_word_recognition', 'linguistic_mappings:de_past_tense_regular_json', '
            unnatural_in_context_learning:dates_unnatural_content', 'natural_instructions:subtask026
            _drop_question_generation', 'hyperbaton', 'symbol_interpretation:plain', 'arithmetic:5
            _digit_addition', 'chess_state_tracking:synthetic_long', 'simp_turing_concept:additional_set_1'
            , 'physics', 'cifar10_classification:base64', 'intersect_geometry:shapes_4', '
            elementary_math_qa:question_with_language_hint', 'goal_step_wikihow:step_inference', '
            unit_conversion:unit_identification', 'natural_instructions:subtask056
            _multirc_classify_correct_answer', 'strange_stories:boolean', 'discourse_marker_prediction', '
            bbq_lite_json:bbq_lite_json_religion_disambig', 'natural_instructions:subtask025
            _cosmosqa_incorrect_answer_generation']"
```
