# OpenReview forum: "How Predictable Are Large Language Model Capabilities? A Case Study on BIG-bench"
_EMNLP/2023/Conference — EMNLP 2023 Findings_

### Official Review · Reviewer_6piE · 2023-08-02

**Typos Grammar Style And Presentation Improvements:** NA
**Soundness:** 3

**Excitement:**

3: Ambivalent: It has merits (e.g., it reports state-of-the-art results, the idea is nice), but there are key weaknesses (e.g., it describes incremental work), and it can significantly benefit from another round of revision. However, I won't object to accepting it if my co-reviewers champion it.

**Missing References:**

NA

**Paper Topic And Main Contributions:**

This paper investigates the extent to which the performance of Large Language Models (LLMs) can be predicted prior to any training. If performance was found to be cheaply predictable, then LLM development could be made more accurate, faster, and cheaper computationally. The authors introduce a novel method for predicting LLM performance based on the experiment records from BIG-bench, which include data from various model families, tasks, parameters, and in-context examples.


The main contribution of this paper lies in the authors’ demonstration of high predictability of LLM performance using an MLP-based predictor, with an R2 score of over 95%, indicating that strong learnable patterns exist within the data. They define the prediction problem using 4 inputs: Model family, number of model parameters (as a proxy for model scale), evaluated tasks, and number of in-context examples. The model’s normalized performance is then predicted. They evaluate many different prediction models, ultimately finding strong performance in Tree-based methods and MLPs. They also provide insight into what test groups (such as zero-shot vs one-shot models) are more difficult to predict, and why. They find certain difficult groups cannot easily make use of multi-group training (such as the zero-shot group) and others are just intrinsically hard to predict (such as models that are larger). They additionally experiment with the intrinsic predictability of certain tasks, and provide hypotheses as to why certain tasks may be more or less predictable than others.


Additionally, the paper introduces the concept of “small-bench,” a condensed subset of BIG-bench tasks, which aims to maintain equivalent informativeness while significantly reducing the computational resources and time required for evaluating new model families. The authors underline the importance of task diversity and value in constructing an informative "small-bench" that can maintain similar predictive power to the larger BIG-bench. They showed that certain methods of creating "small-bench" subsets outperform baseline subsets of BIG-bench that are not explicitly designed for informative prediction.

**Questions For The Authors:**

Is there a reason why certain settings are able to leverage multi-group training more than others? It’s unclear to me why the 0-shot group may not benefit from multi-group training.

In table 1, are the total # of tasks examined the sum of the number of subtasks and the number of tasks? Or are there 313 subtasks that are evaluated which fall into 134 task categories?

How do the small-bench experiments compare to that of BIG-bench? Both in number of tasks and performance? While the subgroups for BIG-bench do seem like more reasonable baselines, is there a specific reason this direct comparison is never made?


**Reasons To Accept:**

The paper introduces a novel method for predicting the performance of Large Language Models (LLMs), which has significant practical implications. This method could aid developers and researchers in conserving resources when training models, given the vast array of possible combinations of models, hyperparameters, evaluation tasks, and in-context examples.


The results of the paper are notable. The high R2  scores indicate that the BIG-bench dataset contains highly predictable patterns, thus validating the authors' approach to performance prediction.


In addition, the authors present the idea of “small-bench,” which could reduce computational barriers for models with limited training computation at their disposal. They provide an understanding of how to construct “small-bench” effectively, suggesting potential applications of these methods to optimize other benchmarks in different contexts

**Reasons To Reject:**

Only RMSE and R2 scores are reported. While this limitation is addressed in the paper, they do not capture the complete picture of a model’s accuracy.

There are a relatively small number of model families in BIG-bench. This could lead to results, even in the challenging Train-Test split, being due to intrinsic correlation between these model families that might not translate to other models.

There is little discussion of the variety of models and tasks in BIG-bench. It’s unclear whether the model families present in BIG-bench are significantly different from each other (at least as much as another model with a different pre-training pipeline might be).


**Reproducibility:**

5: Could easily reproduce the results.

**Reviewer Confidence:**

3: Pretty sure, but there's a chance I missed something. Although I have a good feel for this area in general, I did not carefully check the paper's details, e.g., the math, experimental design, or novelty.

---

> ### Author Rebuttal · Authors · 2023-08-29
>
> Thank you for your detailed review and feedback! We are glad that you think our work offers “significant practical implications”, presents “notable results”, and “could reduce computational barriers [for LLM evaluation]”. In response to your comments, we offer further information and clarification below.
>
> ### Response to reasons to reject
>
> > Only RMSE and R2 scores are reported. While this limitation is addressed in the paper, they do not capture the complete picture of a model’s accuracy.
>
> * In the main paper, we focus on RMSE and R2 scores as they are widely accepted evaluation metrics for regression problems.
> * Additionally we introduced task-average Pearson Correlation and Kendall Rank Correlation in Appendix C.1, in an attempt to get a more comprehensive picture of our prediction model. We reported the full results in Table 4.
> * We will revise our paper to highlight these additional metrics and our considerations.
>
> > There are a relatively small number of model families in BIG-bench. […] It’s unclear whether the model families present in BIG-bench are significantly different from each other.
>
> * We would like to note that the 6 model families in this study are representative, and offer considerable diversity. We provide a brief summary in the following:
>
> | Model Family | Company | Release Date | Pretrain Corpus | #Parameters | Architecture Choices | Infrastructure
> | ----------- | ----------- | ----------- | ----------- | ----------- | ----------- | ----------- |
> | BIG-G (T=0, T=1) | Google  | Jun 2022 | 2.8T tokens | 2M to 137B | Gated-GELU, relative attention | TPUv3, GSPMD |
> | BIG-G Sparse | Google | Jun 2022 | 2.8T tokens | 51M to 46B  | BIG-G + Mixture-of-Experts | TPUv3, GSPMD |
> | PaLM     | Google  | Apr 2022 | 780B tokens | 8B,62B,540B | SwiGLU activation, parallel layers, etc. | TPUv4, Pathway |
> | GPT-3    | OpenAI | May 2020 | 300B tokens | 125M to 175B | Alternating dense and sparse attention | GPUs |
> | Gopher  | DeepMind | Nov 2021 | 300B tokens | 44M to 280B | RMSNorm, relative position encoding | TPUv3, model parallelism, etc. |
>
> * As we chose BIG-bench as our testbed for the study, we have made every effort to incorporate all the model families evaluated in it. Recognizing this limitation, we encourage further community efforts on “broadening observations on LLM capability landscape” (see our discussion on page 8). We hope our methodology presented in this work will remain useful when more observations become available.
>
> > There is little discussion of the variety of models and tasks in BIG-bench.
>
> * Models: Please see the discussion above.
> * Tasks: Quoting from the original BIG-bench paper (https://arxiv.org/abs/2206.04615), “task topics are diverse, drawing problems from linguistics, childhood development, math, common-sense reasoning, biology, physics, social bias, software development, and beyond.” The paper also provides a summary on the task diversity in Fig. 3 and Table App.3. We will include this information and refer our readers to the original paper.
>
> ### Response to questions
>
> > Is there a reason why certain settings are able to leverage multi-group training more than others? It’s unclear to me why the 0-shot group may not benefit from multi-group training.
>
> * For most tasks, there is a huge performance boost when going from zero-shot to one-shot, and the performance improves more stably when more shots become available (see Figure 1 in https://arxiv.org/abs/2102.09690). It is easier to predict 3-shot performance when given {0,1,2}-shot performance, than to predict 0-shot performance when given {1,2,3}-shot performance. This partly explains why the 0-shot group does not benefit much from multi-group training.
>
> > In table 1, are the total # of tasks examined the sum of the number of subtasks and the number of tasks? Or are there 313 subtasks that are evaluated which fall into 134 task categories?
>
> * Sorry for the confusion. There are 313 subtasks that fall into 134 tasks. For example, “arithmetic” is a task, while “2-digit addition” is a subtask under the task of “arithmetic”.
>
> > How do the small-bench experiments compare to that of BIG-bench? Both in number of tasks and performance? While the subgroups for BIG-bench do seem like more reasonable baselines, is there a specific reason this direct comparison is never made?
>
> * To clarify, the “small-bench” problem is a subset search problem. When a new LLM is evaluated on a subset of BIG-bench (i.e., “small-bench”), we aim at maximally recovering the performance on the remaining tasks. Therefore, we compare with different BIG-bench subsets as baselines.
> * To put BIG-bench into comparison directly, BIG-bench will appear at the coordinates (313, 1.0) in Figure 6 in the paper, as it has 313 subtasks in total.

---

### Official Review · Reviewer_WS63 · 2023-08-02

**Soundness:** 5

**Excitement:**

4: Strong: This paper deepens the understanding of some phenomenon or lowers the barriers to an existing research direction.

**Paper Topic And Main Contributions:**

This paper tries to interpolate and extrapolate an LLM's normalized performance metric based on a quadruple of model type, number of parameters, task, and number of shots by training a regression model with existing BIG-Bench statistics. The authors show they could obtain decent RMSE even for a challenging setting, where a tuple of model and task combination is not seen in the training set.

Based on the regression model, the authors further show a method to select a representative subset of BIG-Bench. The subset selected by the proposed approach is more representative than the widely-used BIG-Bench Hard and BIG-Bench Lite.

**Questions For The Authors:**

1. In L473-479, I do not see how the first-layer representation of a MLP could capture diversity and representativeness.

2. There are some issues during the authors' data processing and model building:
   1. In L970-972, the authors remove all performance metrics equal to 0. Despite sparsity in available information to train the model, a useful model should also be able to predict 0 performance metrics. Could authors explain this choice?
   2. The authors do not provide a rationale for choosing specific features when building their most performant model - an MLP-based regressor. For example, why do authors use 6 numerical features for the number of parameters rather than 1?

**Reasons To Accept:**

- The problem the authors are trying to solve is real and very relevant to the needs of LLM evaluation: (1) how to predict an LLM's performance without actually running the experiment, and (2) how to quantitatively select the best subset of comprehensive benchmark for quick evaluation.
- The proposed approach is effective even for challenging scenarios.
- The presentation of the main idea is easy to follow, with comprehensive experimental results to back up.

**Reasons To Reject:**

- Choosing explanatory variables in your design matrix does not have a sufficient explanation. For example, two models with the same architecture could perform widely differently if training one converges while the other fails.
- The authors do not discuss how the emergent abilities of LLMs could influence the regression models' performance. For example, how do the LLMs' emergent abilities contribute to the RMSE of your regression models?
- The authors need to discuss what conditions should be met to apply their proposed approach. For example, the authors do not analyze T0/T5 models (L967-969) because of the unavailability of sufficient data to train the model. However, the authors do not discuss the minimal data requirement to develop a valid analysis with the proposed approach.
- The authors claim that there are limitations of BIG-Bench Hard (BBH) and BIG-Bench Lite (BBL) based on the proposed metrics. However, to establish the discovered BIG-Bench subset's advantage over BBH and BBL, the authors should use an independent source of information. For example, a previously ignored LLM due to low ranking on BBH and HHL is found to rank high on the discovered benchmark, and this model is able to do something that high-ranking models on BBH and BBL could not do well.

**Reproducibility:**

4: Could mostly reproduce the results, but there may be some variation because of sample variance or minor variations in their interpretation of the protocol or method.

**Reviewer Confidence:**

5: Positive that my evaluation is correct. I read the paper very carefully and I am very familiar with related work.

---

> ### Author Rebuttal · Authors · 2023-08-29
>
> We greatly appreciate your thorough review and insightful feedback! It is encouraging to hear that you believe our research questions are “real and very relevant to the needs of LLM evaluation” and our work is backed by “comprehensive experimental results”. We will incorporate your feedback and updated discussions in our next version.
>
> ### Response to reasons to reject
>
> > Choosing explanatory variables in your design matrix does not have a sufficient explanation. For example, two models with the same architecture could perform widely differently if training one converges while the other fails.
>
> * Thanks for raising this point! We understand that LLMs capabilities are dependent on many factors, including pre-training stability and convergence, pre-train corpus composition, etc. In an ideal case, we wish to incorporate all these meta-data as the input feature to our performance prediction model. However, we often don't have access to this information.
> * In this work, we assume that the input features (model_family, n_param) can implicitly learn to capture such information. In the future, we believe our method can be readily expanded to include additional pre-training meta-data when they become available.
>
> > The authors do not discuss how the emergent abilities of LLMs could influence the regression models' performance. For example, how do the LLMs' emergent abilities contribute to the RMSE of your regression models?
>
> * __In general__, performance prediction accuracy on emergent tasks is lower than that on non-emergent tasks. We divide the test set into records associated with emergent and non-emergent tasks (characterized in Appendix E in https://arxiv.org/abs/2206.07682). We take the MLP model trained in the L1 setting, and report RMSE and $R^2$ within each group in the following table.
>
> |  | RMSE ($\downarrow$) | $R^2$ ($\uparrow$) |
> | ----------- | ----------- | ----------- |
> | Emergent | 0.0541 | 93.86% |
>  | Non-emergent | 0.0496 | 95.16% |
>  | All | 0.0499 | 95.07% |
>
> * __In certain cases__, emergence can be predicted by our model accurately. In Figure 7 Subfigure 2, we visualize the predicted and actual performance of BIG-G T=0 models on the three_digit_addition, an emergent task. We hypothesize that emergence on a task may be accurately predicted when a similar emergent task (e.g., similar arithmetic tasks) appears in the training set. See Line 322-334 and Line 1065-1075 for more discussion.
>
> > The authors need to discuss what conditions should be met to apply their proposed approach. For example, the authors do not analyze T0/T5 models (L967-969) because of the unavailability of sufficient data to train the model. However, the authors do not discuss the minimal data requirement to develop a valid analysis with the proposed approach.
>
> * T0/T5 models have 14 experiment records in BIG-bench. In contrast, BIG-G/PaLM/GPT-3 models have 5000+ records. Due to the magnitude of difference, we decided to discard T0/T5 models in our experiments.
> * Regarding the minimal data requirement, we think this is highly related with the “small-bench” problem we are trying to solve in Sec. 5.
>
> > The authors claim that there are limitations of BIG-Bench Hard (BBH) and BIG-Bench Lite (BBL) based on the proposed metrics. However, to establish the discovered BIG-Bench subset's advantage over BBH and BBL, the authors should use an independent source of information. For example, a previously ignored LLM due to low ranking on BBH and BBL is found to rank high on the discovered benchmark, and this model is able to do something that high-ranking models on BBH and BBL could not do well.
>
> * Thank you for your suggestions! We agree it would be more convincing if we can validate the small-bench candidates on more model families (i.e., using independent sources of information). However this requires running new LLMs on the hundreds of tasks in BIG-bench, which is prohibitive for us at this time.
> * During the rebuttal period, we examined the existing model families in our study and indeed found several cases where BBH fails to represent the full BIG-bench. For example, according to BBH, BIG-G T=1 2B is more capable than GPT-3 Large on the full BIG-bench; however, in reality, GPT-3 Large is better than BIG-G T=1 2B, which is captured more accurately when a small-bench candidate is used.
>
> |  | BIG-G T=1 2B Wins | Tie | GPT-3 Large Wins |
> | ----------- | ----------- | ----------- | ----------- |
> | Performance Recovered from BBH (24 subtasks) | 20.1% | 70.1% | 9.8% |
> | Performance Recovered from small-bench (24 subtasks) | 19.1% | 56.3% | 24.6% |
> | Ground-truth full BIG-bench Performance | 14.3% | 55.3% | 30.4% |
>
> ### Response to questions
>
> > In L473-479, I do not see how the first-layer representation of a MLP could capture diversity and representativeness.
>
> * The first-layer representation of MLP can be considered as representation for the task. We manually verified this by visualizing the representations in 2D space (e.g., similar tasks are close to each other in the first-layer representation space).
> * Diversity is captured by clustering these representations. We select one task from each cluster, so that similar tasks will be deduplicated, and this promotes task diversity.
>
> > In L970-972, the authors remove all performance metrics equal to 0. Despite sparsity in available information to train the model, a useful model should also be able to predict 0 performance metrics. Could authors explain this choice?
>
> * To clarify, we remove subtasks that are _too hard_ so that _no_ model achieves performance better than 0. We believe these are outlier tasks and provide little information in training the performance prediction model.
>
> > The authors do not provide a rationale for choosing specific features when building their most performant model - an MLP-based regressor. For example, why do authors use 6 numerical features for the number of parameters rather than 1?
>
> * We use 6 numerical features mainly because the BIG-bench repository directly provides 3 features (# total parameters, # non-embedding parameters, # FLOP-matched non-embedding parameters). Further, since scaling laws typically use the log of the pre-train scale as a feature, we take the log of these 3 values.
> * It is possible that alternative featurizations may lead to better performance. However due to the complexity of our study we did not further investigate this.

---

### Official Review · Reviewer_Ku3n · 2023-08-10

**Soundness:** 2

**Excitement:**

3: Ambivalent: It has merits (e.g., it reports state-of-the-art results, the idea is nice), but there are key weaknesses (e.g., it describes incremental work), and it can significantly benefit from another round of revision. However, I won't object to accepting it if my co-reviewers champion it.

**Paper Topic And Main Contributions:**

The theme of this paper is the predictability of the capabilities of Large Language Models (LLM). Specifically, it focuses on using five types of data as a dataset: the model family, the number of parameters, the type of small tasks, the number of context examples, and the performance results on sub-tasks. An additional model was employed to learn from this dataset. Based on this approach, experiments were conducted, high-quality data subsets were sought and organized, and findings and analyses from the experiments were summarized.

**Reasons To Accept:**

1. The experiments in this article demonstrate that some capabilities of LLMs are predictable. This method might serve as a reference metric for future LLM development.
2. Furthermore, the article emphasizes the significant impact of data on LLMs and suggests a vast space for further research.

**Reasons To Reject:**

1. This article relies solely on superficial information to assess the predictability of LLM capabilities, which weakens its persuasiveness.
2. The experiments in this article are quite limited, as all tests were conducted on just one dataset. This might introduce potential biases and uncertainties regarding its generalization capabilities.

**Reproducibility:**

4: Could mostly reproduce the results, but there may be some variation because of sample variance or minor variations in their interpretation of the protocol or method.

**Reviewer Confidence:**

3: Pretty sure, but there's a chance I missed something. Although I have a good feel for this area in general, I did not carefully check the paper's details, e.g., the math, experimental design, or novelty.

---

> ### Author Rebuttal · Authors · 2023-08-29
>
> Thank you for your comments! It appears that there are some misunderstandings that we wish to clarify. We hope the following response helps address your concerns.
>
> ### Response to reasons to reject
>
> > This article relies solely on superficial information to assess the predictability of LLM capabilities, which weakens its persuasiveness.
>
> * We are not entirely sure what this statement is referring to. The experiment records used in this study are real evaluation of LLMs on diverse language tasks. We would appreciate it if reviewer Ku3n can provide more information regarding the “superficialness”.
> * If “superficial” refers to sourcing the experiment records from a public repository, we consider this as a strength rather than a limitation, as these records are open to the public and it is easy to reproduce our study.
>
> > The experiments in this article are quite limited, as all tests were conducted on just one dataset. This might introduce potential biases and uncertainties regarding its generalization capabilities.
>
> * Our work is a meta-analysis on 313 (sub-)tasks and 51 language models presented in BIG-bench. BIG-bench is one of the most comprehensive evaluation benchmarks available for large language models. We believe this represents our most rigorous effort to address "generalization" across various models and tasks.
> * In addition to the default experiment (Sec. 3), we have experiments on controlling training set composition (Sec. 3.4); constructing more challenging train-test splits (Sec. 4); designing “small-bench” using the performance prediction model (Sec. 5). We hope this addresses your concern that the experiments are limited.
>
> ### Reproducibility
>
> > Would be hard pressed to reproduce the results. The contribution depends on data that are simply not available outside the author's institution or consortium; not enough details are provided.
>
> * The experiment records we use in this study come from the BIG-bench repository (https://github.com/google/BIG-bench) which is open to the public.
> * We have included our code in the supplementary material. Additionally, we provide dataset creation and filtering details in Appendix A and model training details such as hyperparameters, hardware information in Appendix D.

---

### Meta-Review · Area_Chair_4AHn · 2023-09-12

**Recommendation:** 4

**Metareview:**

The main conclusions of the reviews and the post-rebuttal discussions:
- 2/ 3 reviewers consider the paper sound (scores 2, 5, 3)
- 3/ 3 reviewers find the paper exciting (scores 4, 3, 3)

From reading the rebuttal and seeing the scores above, I find that the reviewers consider strong points for soundness the following:
- tackle an important problem: (1) how to predict an LLM's performance without actually running the experiment, and (2) how to quantitatively select the best subset of comprehensive benchmark for quick evaluation.
- the authors present the idea of “small-bench,” which could reduce computational barriers for models with limited training computation
- solution is effective even for challenging scenarios.
- comprehensive experimental results
- easy to read

---

### Decision · Program_Chairs · 2023-10-07

**Decision:**

Accept-Findings

**Comment:**

The main conclusions of the reviews and the post-rebuttal discussions:
- 2/ 3 reviewers consider the paper sound (scores 2, 5, 3)
- 3/ 3 reviewers find the paper exciting (scores 4, 3, 3)

From reading the rebuttal and seeing the scores above, I find that the reviewers consider strong points for soundness the following:
- tackle an important problem: (1) how to predict an LLM's performance without actually running the experiment, and (2) how to quantitatively select the best subset of comprehensive benchmark for quick evaluation.
- the authors present the idea of “small-bench,” which could reduce computational barriers for models with limited training computation
- solution is effective even for challenging scenarios.
- comprehensive experimental results
- easy to read